ns**nature communications**

# CryoEM and computational modeling structural insights into the pH regulator NBCn1

Weiguang Wang [1,2,6], Hristina R. Zhekova [3,6], Kirill Tsirulnikov[1], B. Sridhar Dwadasi [3], Ehecatl Guzman Aparicio [3], Rustam Azimov[1], Natalia Abuladze[1], Liyo Kao[1], Dora Acuna [1], D. Peter Tieleman [3], Z. Hong Zhou [2,4], Alexander Pushkin[1] & Ira Kurtz [1,5] ✉

Breast cancer cells survive despite being exposed to a toxic acidic extracellular environment, by utilizing the NBCn1 transporter. The molecular basis for this phenomenon is unknown, given the lack of an NBCn1 atomic structural model. We therefore determined the 3.3 Å cryoEM structure of the human NBCn1 outward facing (OF) conformational state with densities corresponding to the transported ions in the ion coordination site. We further generated NBCn1 inward facing (IF) and intermediate (occluded) structures and characterized the transport cycle and the ion dynamics in the IF and OF states. The results showed that NBCn1 utilizes an elevator-type transport mechanism with a small vertical shift of the ion coordination site between OF and IF conformational states and that the transported ions permeate without significant energy barriers. Functional experiments showed that NBCn1 has an extremely high ion turnover rate (TOR) of ~15,000 s$^{-1}$. The unusually high NBCn1 TOR value associated with the small protein structural changes during the OF to IF transitions and the favorable ion permeation energetics provides breast cancer cells with a highly efficient base loading mechanism contributing to their survival advantage.

Tumor cell development is profoundly affected by the acidity of the extracellular and intracellular environment[1–4]. As a result of increased aerobic lactic acid production (Warburg effect), cancer cells are exposed to an acidic local pH that promotes genetic instability[5,6], precancerous histologic changes[7,8], invasive metastatic tumor growth[9,10] and drug resistance[11]. An increase in extracellular acidity also stimulates the expression of both matrix-degrading enzymes and angiogenic factors that result in an increase in local invasive and metastatic tumor cell potential[12–14]. As an adaptive response to survive in this acidic environment, tumors have developed various compensatory mechanisms that stabilize or increase their intracellular pH (pH$_i$)[1,3,4,15–18]. Among these protective adaptations that lead to an increase in pH$_i$, are an upregulation of plasma membrane transport proteins, which mediate cellular base loading or H$^+$ efflux[1,2,4,7]. Not only is tumor cell survival enhanced, but the relatively alkaline pH$_i$ further promotes cancer cell proliferation and growth[2,7,19].

Throughout the world, breast cancer is the most common cancer in women and is also the most common cause of cancer death[20,21]. Like

[1]Department of Medicine, Division of Nephrology, David Geffen School of Medicine, University of California, Los Angeles, CA, USA. [2]Electron Imaging Center for Nanomachines, California NanoSystems Institute, University of California, Los Angeles, CA, USA. [3]Centre for Molecular Simulation, Department of Biological Sciences, University of Calgary, Calgary, Canada. [4]Department of Microbiology, Immunology and Molecular Genetics, University of California, Los Angeles, CA, USA. [5]Brain Research Institute, University of California, Los Angeles, CA, USA. [6]These authors contributed equally: Weiguang Wang, Hristina R. Zhekova. ✉e-mail: ikurtz@mednet.ucla.edu

other tumors, breast cancers have developed adaptive mechanisms to survive in the presence of an acidic extracellular local environment[22]. Systems analyses in breast cancer cells have shown a coupling between an increase in lactic acid production and decreased oxygen uptake, with the ability to elevate the $pH_i$ and prevent its decline thereby prolonging tumor survival[3]. Accordingly, lowering $pH_i$ in breast cancer cells is a potential therapeutic approach that can enhance the response to inhibitors of tumor cell growth[3,7].

A major plasma membrane ion transport protein in breast tumors, which was identified by functional genomics analyses, is NBCn1 (SLC4A7), that loads base equivalents into cells thereby elevating $pH_i$ while acidifying the extracellular environment[23–25]. NBCn1 belongs to the SLC4 protein family whose members mediate in general transport of $HCO_3^-$ or $CO_3^{2-}$ coupled to $Na^+$ and/or $Cl^-$ in an exchange or symport mode[26]. Importantly, previous genome-wide association studies have established a connection between variations in SLC4A7 SNPs and the risk of developing breast cancer[27–30]. Furthermore, SLC4A7 mRNA expression levels particularly in luminal A or basal-like/triple-negative breast cancers have been identified as correlating with patient survival[31]. In murine models, loss of NBCn1 impairs breast cancer cell growth[23], whereas an increase in NBCn1 expression due to loss of RPTPγ is associated with malignant transformation of normal breast tissue, decreased survival and more rapid recurrence[32]. In MCF-7 breast cancer cells, NBCn1 plays an important role in cell cycle progression and entry into mitosis[33,34]. A recent study has confirmed that targeting NBCn1 with monoclonal antibodies results in breast cancer cell death in vitro[35].

Here we determined the near atomic resolution structure by cryogenic electron microscopy (cryoEM) of human NBCn1 with densities corresponding to the transported ions and computationally modeled the conformational state changes during its transport cycle. Small protein structural changes during its elevator OF ↔ IF transitions are associated with an extremely high measured ion turnover rate (TOR) and base flux. By utilizing this highly efficient base loading transporter, breast cancer cells can have a survival advantage.

## Results

### CryoEM structure of full-length human NBCn1

Full-length human NBCn1-A (NP_003606.3), expressed in HEK293 cells and purified by affinity and size exclusion chromatography, was immediately used for cryoEM grid preparation. Data processing performed in RELION[36] and cryoSPARC[37], yielded a homodimer (state 1; 3.5 Å) with both monomers having 14 transmembrane helices (TMs) (Fig. 1a–f, Supplementary Figs. S1a–e, S2). Other dimeric states had one (mixed dimer state 2; 3.4 Å) or both (homodimeric state 3; 3.3 Å) monomers lacking resolved TMs 13 and 14 because of their flexibility (Supplementary Fig. S1a, d, e; Supplementary Table 1). NBCn1 consists of TMs 1-14, nine amphipathic helices (H1-9) and a terminal helix located in the cytoplasmic C-terminus (H10) (Fig. 1a–c; Supplementary Fig. S2). The arrangement of TMs 1-7 and TMs 8-14 mirror each other as inverted repeats. The fourteen transmembrane helices are subdivided into two domains: the gate domain and the core domain (Fig. 1d), which are demarcated by a gap between the two domains. TMs 1-4 and TMs 8-11, along with H1-2 and H7, constitute the core domain, whereas TMs 5-7 and TMs 12-14, along with H8-9, establish the gate domain. H1 located intracellularly, runs parallel to the cell membrane before entering the membrane. The connections of loop–H2–loop (intracellular loop 2; IL2) and loop–H7–loop (extracellular loop 4; EL4) span the core and gate domains on the cytoplasmic and extracellular sides of the plasma membrane, respectively (Fig. 1c), thereby bridging the core domain with the gate domain. In addition, the NBCn1 structure features a notably large EL3. The assembly of EL3 is composed of four short α-helices (H3 to H6) and a single pair of anti-parallel β-strands, together forming a relatively rigid domain-like configuration. The EL3 structure is stabilized by hydrogen bonds between the anti-parallel β-strands and two intramolecular disulfide bonds, Cys766–Cys814 and

Cys768–Cys802. Three asparagine residues, Asn776, Asn786, and Asn796, have densities indicating the presence of covalently bound N-acetylglucosamine sugars (Fig. 1b, c, f) unlike other SLC4 $Na^+$-coupled SLC4 transporters possessing two glycosylation sites[38,39].

### NBCn1 dimerization and intramolecular interactions

NBCn1 dimerization is primarily stabilized by hydrophobic interactions between the TM6 helices of the dimeric subunits (Supplementary Fig. S3a–d). Key roles are played by residues at the dimeric interface, which form hydrophobic pairs that include Pro822, Phe826, Ile830, Phe833, Phe836, Phe837, and Phe841 (Supplementary Fig. S3c). π-π stacking interactions between the face-to-face aromatic rings of phenylalanine are also involved. Dimerization of EL3 is facilitated by the hydrogen bonds between the hydroxyl oxygen of Thr762 and the amidogen nitrogen of Thr798 from the neighboring subunit; and between the backbone carbonyl group of Asp759 and the amidogen nitrogen of Thr798 from the neighboring subunit (Supplementary Fig. S3a). The EL3 structure is stabilized by two intramolecular disulfide bonds, Cys766–Cys814 and Cys768–Cys802, as well as hydrogen bonds between the anti-parallel β-strand (Supplementary Fig. S3b). IL5 may interact electrostatically with IL3 and IL6 from the neighboring monomer (Supplementary Fig. S3d; Supplementary Movie 1).

### NBCn1 OF ion coordination site

TM1, TM8, TM3, and TM10, and the antiparallel β-sheets preceding the TMs 3 and 10 in the core domain, together with TM5 in the gate domain, contribute to ion coordination (Figs. 1e, 2a–c). In the surface-rendered NBCn1 atomic model, these transmembrane helices present as an extracellular-facing cavity (Fig. 2a). Solvent accesses to the ion coordination area from the extracellular side of the transporter indicate that this is an OF conformational state. Within the ~2.7 Å resolution cryoEM map of the ion coordination area (Supplementary Fig. S1d), three densities were detected approximately halfway across the transporter in the central $S1^{cryoEM}$(OF) ion binding site (ion coordination site (Fig. 2b, c). The cryoEM map of the site visualized at different contour levels and its surrounding residues (Supplementary Fig. S4) demonstrates that the densities likely correspond to two $Na^+$ and a $CO_3^{2-}$ ion. The carbonyl group of Ala975, hydroxyl group of Thr977, and the carboxyl group of Asp930 coordinate the first $Na^+$ ion ($Na^+_{1.1}$) with 3.3 Å, 3.6 Å and 2.9 Å distances, respectively. A second $Na^+$ ion ($Na^+_{1.2}$) is coordinated by the carbonyl oxygen of Ile731 and two $CO_3^{2-}$ oxygens. The $CO_3^{2-}$ ion is located 3.8 Å from $Na^+_{1.1}$. One $CO_3^{2-}$ oxygen is coordinated with both the backbone amide group and the side chain hydroxyl group of Thr668 with distances of 4.0 Å and 2.6 Å, respectively. A second ion binding site, $S2^{cryoEM}$(OF), in the OF ion permeation pathway also contained three densities (Fig. 2b, c; Supplementary Fig. S4) likely corresponding to two $Na^+$ and a $CO_3^{2-}$ ion. The first $Na^+_{2.1}$ was coordinated by the carboxyl group of Glu738 and the amino group of Lys742 at 3.6 Å and 4.2 Å distances, respectively. The second $Na^+_{2.2}$ was coordinated by the carbonyl oxygens of Cys619 and Leu665 at distances of 3.3 Å and 4.2 Å, respectively. The second $CO_3^{2-}$ anion interacts with $Na^+_{2.1}$ and $Na^+_{2.2}$ at distances of 4.1 Å and 5 Å, respectively.

### IF homology model of NBCn1

We generated an IF homology model of NBCn1 (Supplementary Fig. S5a) using as a template the recently published cryoEM IF structure of AE2[40], a homologous SLC4 anion exchanger, using the Swiss-Model server[41]. The superposed IF and OF NBCn1 states are presented in Supplementary Fig. S5b. Overlap of the OF and IF NBCn1 conformational states shows a small vertical displacement of the core domain with respect to the rigid gate domain, which moves vertically the position of the $S1^{MD}$ site (~5 Å) during the OF to IF transition (Fig. 3, Supplementary Fig. S5b). The conformational transition is accompanied by a significant movement of IL5 (Supplementary Movie 1). Separate alignments of the gate and core domains in both states show

 

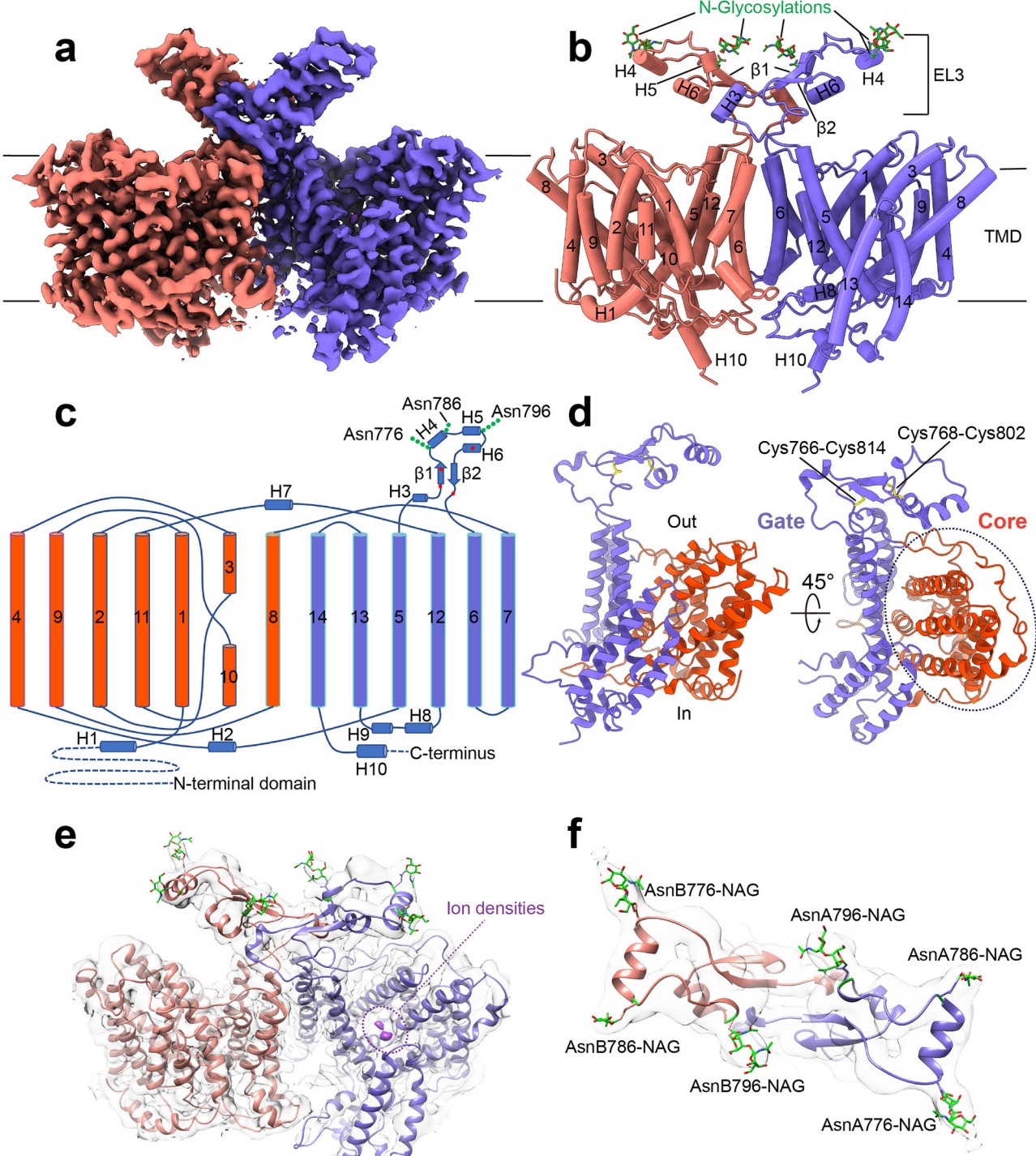

**Fig. 1 | CryoEM structure of NBCn1. a** CryoEM density map of the NBCn1 dimer with the monomers colored in violet and pink. **b** Atomic model of the NBCn1 dimer in the same orientation as in (**a**). TMs 1-14 and helices H1-10 are shown as cylinders. The N-glycosylations (NAG) are shown in green and labeled accordingly. **c** Topology and domain arrangement of the NBCn1 monomer. Four highly conserved cysteines are shown as red dots. The branched structures at Asn776, Asn786, and Asn796 represent N-linked glycosylation. **d** Ribbon diagram of the atomic structure of an NBCn1 monomer with the gate (violet) and core (pink) domains. Side (left) and top (right) views are shown. The regions involved in the formation of the two disulfide bonds are highlighted in yellow. **e** CryoEM densities corresponding to ions are shown in purple. The atomic model is overlaid with the cryoEM map. **f** CryoEM densities corresponding to N-glycosylations at Asn776, Asn786 and Asn796 are shown. The atomic model is overlaid with the cryoEM map.

that the overall transmembrane helices remain relatively unchanged during the OF to IF transition, with both gate and core domain RMSD values being smaller than 1 Å (Supplementary Fig. S5c). The NBCn1 OF to IF conformational change represents an elevator-type mechanism involving vertical movement of the core domain with respect to the gate domain without major structural reorganization.

## Ion permeation dynamics in the NBCn1 OF and IF conformations from SILCS and unrestrained 1 μs MD simulations

To probe the behavior of anions and cations in the NBCn1 structures, we employed the Site Identification by Ligand Competitive Saturation (SILCS) method[42]. Figure 3a displays the SILCS cationic and anionic density maps. Both the OF and IF conformations feature well-defined

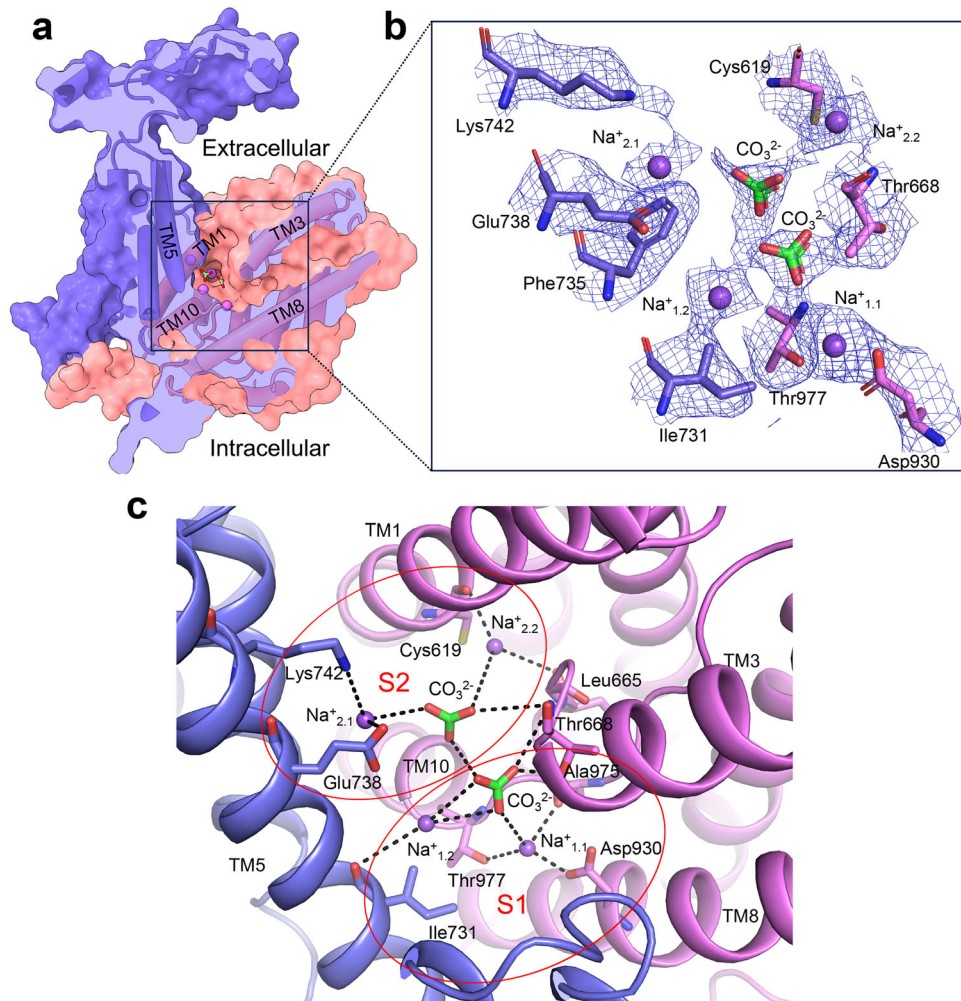

**Fig. 2 | Ion binding sites in the NBCn1 OF conformation. a** Surface-rendered NBCn1 monomer model showing an extracellular-facing cavity. $Na^+$ and $CO_3^{2-}$ ions are shown as purple spheres and green-red sticks, respectively. **b** Magnified view of the cryoEM densities of $Na^+$, $CO_3^{2-}$ and surrounding residues at the same contour level of 0.34σ. **c** The $Na^+$ and $CO_3^{2-}$ ions coordinated at TM1, TM3, TM5, TM8 and TM10 in $S1^{cryoEM}(OF)$ and $S2^{cryoEM}(OF)$ sites.

permeation pathways for cations and anions. In the OF conformation, the cationic permeation pathway is along the second half of TM5, whereas the anionic pathway is along TM3 lined by acidic and basic residues, respectively. In the IF conformational state, the anionic pathway includes a cluster of positively charged residues at the intracellular hairpin junction between TM6 and TM7, while the cationic pathway is located along negatively charged residues from H10 and the first half of TM5. In both conformational states, a sizeable cationic density is observed in the area of the $S1^{MD}$ site Asp930 residue, which has been identified as coordinating $Na^+_{1.1}$ (Fig. 2c). In addition to SILCS maps, to assess ion dynamics in NBCn1 using physiological relevant ions, we generated average ion density maps from unrestrained 1 μs molecular dynamics (MD) simulations of apo-NBCn1 in OF and IF conformations (three replicas per state) in a solution containing 75 mM NaCl, 37.5 mM NaHCO₃ and 37.5 mM Na₂CO₃. The MD ion maps show similar permeation behavior of the cations and anions as the SILCS simulations (Fig. 3b).

**Position-restrained and unrestrained 200 ns MD simulations of NBCn1 for identification and refinement of ion binding sites in the OF and IF conformational states**

To identify putative ion binding sites within the permeation pathways and in the vicinity of the protein center, a series of simulation replicas were prepared, where a $2Na^+$-$CO_3^{2-}$ ion load was moved by 0.3 Å along the z-axis, away from $S1^{cryoEM}(OF)$ within the OF and IF permeation

cavities. This resulted in 71 replicas per conformational state, which were submitted to 100 ns (per replica) simulations with restraints imposed on the C atom of the $CO_3^{2-}$ ion that prohibited the anion motion along the z-axis and allowed its motion only within the XY plane, thereby ensuring exhaustive sampling of the OF and IF permeation cavities without biasing along a predefined linear pathway. Figure 3c shows the cumulative anion density plots (green mesh) of all replicas in the OF and IF conformational states and the free energy diagrams based on the anion density for the $2Na^+$-$CO_3^{2-}$ load, which reveal several shallow minima (i.e. putative transient anion binding sites) along the anion permeation pathways. The prominent minima closest to the protein center are labeled $S1^{MD}(OF)$, $S2^{MD}(OF)$, $S1^{MD}(IF)$, and $S2^{MD}(IF)$ and correspond to either the central ion binding sites in the OF and IF state ($S1^{MD}(OF)$ and $S1^{MD}(IF)$) respectively) or binding sites distal to this site in the OF and IF conformation permeation pathways ($S2^{MD}(OF)$ and $S2^{MD}(IF)$ respectively). In the OF state the anion moves in the OF cavity, overcoming a small barrier of -1.5 kcal·mol⁻¹ between the $S1^{MD}(OF)$ and $S2^{MD}(OF)$ site. It should be noted that the $S2^{MD}(OF)$ site and the remaining energy minima lining the OF conformation anion permeation path are separated by small energy barriers -1–1.5 kcal·mol⁻¹ leading to essentially almost unobstructed permeation of the anions within the OF permeation cavity. In the IF state, the anion permeation has both a lateral and a horizontal component (as seen in the SILCS and MD maps, Fig. 3a, b) with a higher

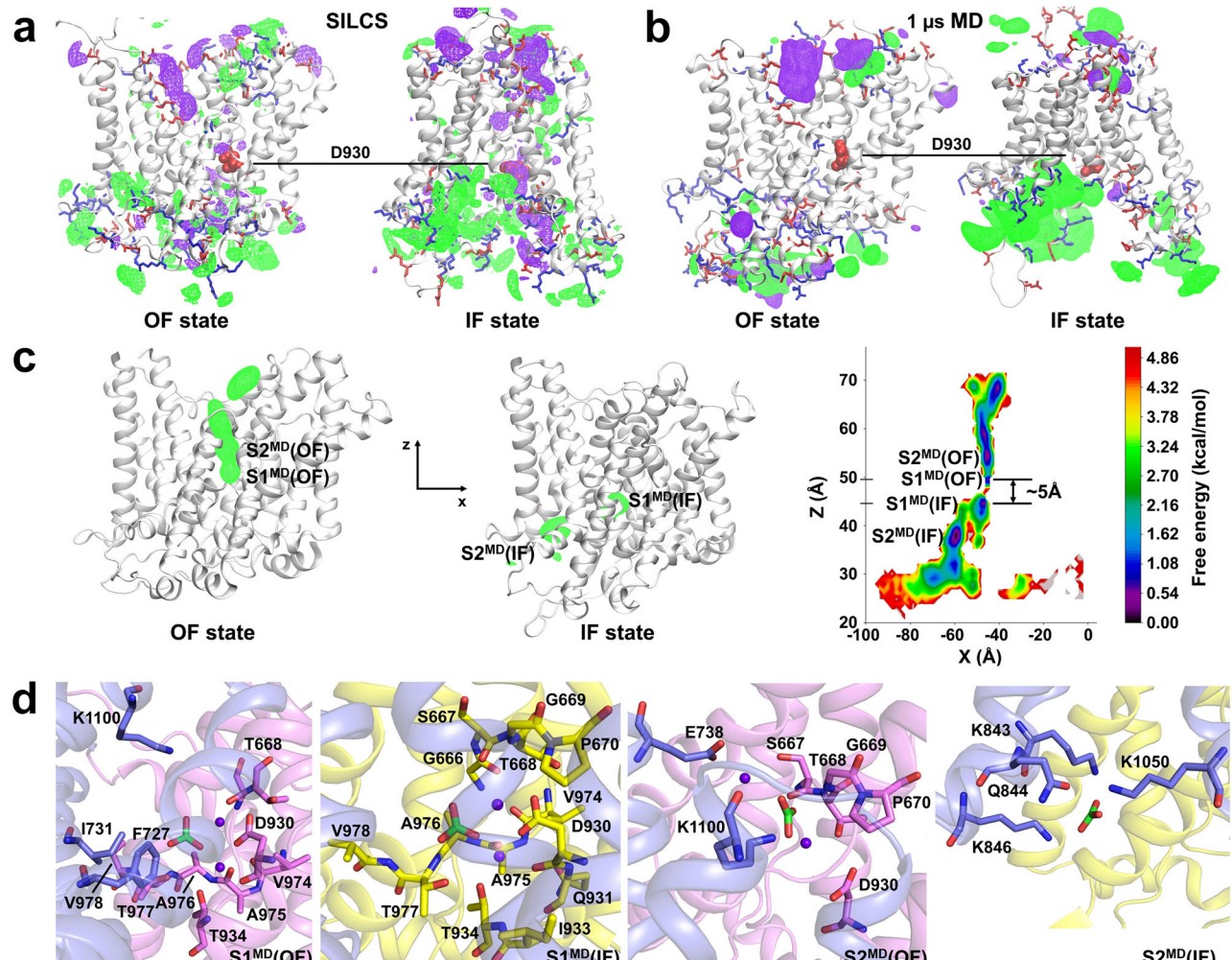

**Fig. 3 | Computational modeling of apo-NBCn1 and NBCn1 with a 2Na⁺·CO₃²⁻ ion load.** The long EL3 is omitted for clarity. **a** SILCS maps of OF and IF NBCn1. Cation and anion densities are presented as purple and green mesh respectively. The acidic and basic residues lining the OF and IF cavities are shown as red and blue sticks, respectively. The central residue D930 is shown as red spheres. **b** Average maps from 1 μs MD simulations. Na⁺ and CO₃²⁻ densities are presented as purple and green mesh respectively. The acidic and basic residues lining the OF and IF cavities are shown as red and blue sticks, respectively. The central residue D930 is shown as red spheres. **c** OF (left) and IF (middle) state with their corresponding cumulative CO₃²⁻ densities (green mesh) calculated from series of MD simulations with position restraints on the C-atom of CO₃²⁻, which prevents it from moving along the z-axis. The coordinate system used in the simulations is also shown. Free energy map (right) projected in the xz-plane and corresponding to the CO₃²⁻ densities in the OF and IF state. Four putative binding sites are designated on the map from the four minima closest to the center of the protein. The ~5 Å vertical displacement of the S1^MD site is shown as well. **d** Amino acid composition of the four binding sites identified in (**c**). The color coding is as follows: Gate/gate residues (purple helices/sticks), core/core residues (pink helices/sticks in the OF state and yellow helices/sticks in the IF state), CO₃²⁻ (green sticks), Na⁺ purple spheres.

barrier of ~2.2 kcal·mol⁻¹ between the S1^MD(IF) and S2^MD(IF) minima. The multiple positively charged residues in the area of site S2^MD(IF) pull the anion toward them and aid in the overcoming of the barrier. Similarly to the anion permeation pathway in the OF state, between site S2^MD(IF) and the IF cavity exit, there are additional minima divided by small barriers (~1.5 kcal·mol⁻¹) that allow the anions to easily exit the IF cavity once they reach site S2^MD(IF).

To refine further the major transport binding sites for Na⁺ and CO₃²⁻ coordination in the vicinity of the protein center and to assess whether the 2Na⁺·CO₃²⁻ ion load remains bound to the central sites without position restraints, we performed a series of 200 ns unrestrained MD simulations of OF and IF NBCn1 loaded with a 2Na⁺·CO₃²⁻ ion load, where the ions were placed initially at the positions of Na⁺_{1.1}, Na⁺_{1.2}, and CO₃²⁻ of the S1^cryoEM(OF) site. Ion time series for these simulations are presented in Supplementary Fig. S6 and show prolonged binding of the ions of the 2Na⁺·CO₃²⁻ ion load in the vicinity of the protein center (sites S1^MD(OF) and S2^MD(OF)) in the NBCn1 OF state.

In the IF state, the 2Na⁺·CO₃²⁻ ion load leads to prolonged anion binding at the protein center and at the anion entry to the IF cavity (sites S1^MD(IF) and S2^MD(IF)).

In the OF conformation, the ions remain localized at the S1^MD(OF) site before eventually dissociating, passing transiently through S2^MD(OF) site and leaving the OF permeation cavity together. The S1^MD sites involve a stretch of residues from the N-terminus of TM10 (Val974-Val978), which coordinate CO₃²⁻ via backbone N-H bonds, the Asp930 and Thr934 residues from TM8 (which are the main Na⁺ coordinating residues), residue Thr668 from TM3 and residues Phe727, Ile731, and Lys1100 from the gate helices TM5 and TM13 (Fig. 3d). During the OF to IF conformational change, the S1^MD(OF) site is translocated vertically ~5 Å and transforms into the S1^MD(IF) site, where most of the coordinating residues remain the same as in the OF conformation. Important differences are the loss of coordination by the gate domain residues Phe727, Ile731, and Lys1100 in S1^MD(IF) and the engagement of more residues from the N-terminus of TM3

(Gly666-Pro670) in ion coordination. The S2$^{MD}$(OF) site consists of residues Glu738 and Lys1100 from the gate domain, residues Ser667-Pro670 from TM3 and residue Asp930 from TM8 of the core domain (Fig. 3d). In the IF state, the S2$^{MD}$(IF) site involves gate domain residues Lys843, Gln844, Lys846, and Lys1050 positioned at the dimerization interface, at the IF anion permeation pathway resolved from our SILCS and MD simulations (Fig. 3a, b).

### Absolute ion binding free energy in S1$^{MD}$(OF) and S1$^{MD}$(IF)

Absolute free energy of binding (ΔG) calculations were conducted in the OF and IF conformational states with 2Na$^+$-CO$_3^{2-}$ and Na$^+$-HCO$_3^-$ ion loads in the S1$^{MD}$(OF/IF) sites identified from the position-restrained and unrestrained MD simulations. The combination of two Na$^+$ and a CO$_3^{2-}$ yielded significantly more negative ΔG values than the combination of one Na$^+$ and a HCO$_3^-$ (−72.30 vs −2.98 kcal·mol$^{-1}$ for OF NBCn1 and −96.81 vs −10.94 kcal·mol$^{-1}$ for IF NBCn1, respectively), indicating stronger stabilization of 2Na$^+$-CO$_3^{2-}$ than Na$^+$-HCO$_3^-$ in the NBCn1 ion binding site compared to the same ions in a water box (Supplementary Table 2). These findings suggest that CO$_3^{2-}$ is the preferred NBCn1 substrate.

### Computational modeling of the OF ↔ IF transition and identification of an occluded apo intermediate state in the transport cycle

To identify an occluded intermediate state we generated 103 intermediate apo-NBCn1 conformations, which sample the space between the IF and OF states using Climber[43]. A subset of 16 of these intermediate conformations was subjected to 1 μs long unrestrained MD simulations. After clustering of all simulated trajectories in the space of the "extracellular distance" (measured between the centers of mass of TM3 and the gate domain) and the "intracellular distance" (measured between the centers of mass of TM10 and the gate domain), three clusters corresponding to the OF and IF conformational states and an occluded apo intermediate (Int) state were obtained. The free energy map corresponding to the amassed trajectories with position of the OF, IF, and Int cluster centers is shown on Fig. 4 together with representative OF, IF, and Int structures from these simulations. The three conformational states form large basins in the free energy graph on Fig. 4 separated by small energy barriers. The Int conformational state is of slightly higher energy than the OF and IF states (with ~0.5–1 kcal·mol$^{-1}$) and likely represents the apo-int conformation during the IF to OF empty transporter transition. Water maps calculated from the clusters corresponding to the three conformational states are shown in Supplementary Fig. S7 (dark blue mesh) and demonstrate the existence of alternating hydrophobic contacts (Supplementary Fig. S8) formed by the gate (TMs 5 and 12) and core (TMs 3, 8, and 10) domain helices that prevent water permeation between the extracellular and intracellular solutions. Starting from the IF conformational state, after the Na$^+$ and CO$_3^{2-}$ ions are released into the cytoplasm, the upward motion of the core domain shifts TMs 1, 3, and 8 upwards and TM10 upwards and forwards. This shift first leads to occlusion of the IF cavity, while the OF cavity is still occluded by interface hydrophobic residue contacts resulting in the occluded apo-Int state (Supplementary Fig. S7, middle). Further upward motion of the core domain opens a wide OF cavity, which allows immediate access of the extracellular solution to the residues in the S1 site. Thus, during the IF to OF transition, the core residues of the S1$^{MD}$ site are alternately exposed to the intracellular and extracellular solutions via the IF and OF permeation cavities (lined by core domain residues in TMs 1, 3, 8, and 10 and gate domain residues in TMs 5, 12, 13, and 14). In the Int state, a visible water density in the area of the S1$^{MD}$ site binding residues that is isolated both from the extracellular and intracellular solutions, indicates that the binding site is not entirely dehydrated in the occluded apo state. Importantly, the TMs 1, 3, 8, and 10 from the core in the Int conformational state assume an intermediate position between those of

the IF and OF conformational states (Fig. 4, Supplementary Fig. S7), which leads to occlusion both at the intracellular and extracellular sides of the transporter.

### NBCn1 transport studies

Cysteine substitution of key residues in the ion-binding regions and permeation pathways of the OF and IF states was performed to probe the functional importance of these regions. Figure 5 shows NBCn1-H (NP_001308036.1) Na$^+$-driven base transport corrected for cell-surface plasma membrane (PM) expression (Supplementary Fig. S9). In general, cysteine mutations of residues in S1$^{cryoEM}$(OF), S2$^{cryoEM}$(OF), S1$^{MD}$(OF/IF), S2$^{MD}$(OF/IF) sites and the ion permeation pathways determined by cryoEM, SILCS, and MD simulations significantly affected NBCn1 transport (Figs. 3, 5 and Supplementary Fig. S10). The NBCn1 TOR number was calculated (Supplementary Table 3) from the measured wt-NBCn1 transport flux and cell membrane monomer number (Supplementary Fig. S11). Given a wt-NBCn1 flux of 8.78 × 10$^{-16}$ (mole·cell$^{-1}$·s$^{-1}$) and a cell plasma membrane monomer expression of 5.77 × 10$^{-20}$ (mole·cell$^{-1}$), the calculated NBCn1 TOR value was therefore ~ 15,000 (s$^{-1}$) (Supplementary Table 3). This extremely high TOR number is in keeping with other SLC4 transporters that share this property[44] exceeding the TOR values of all other transporters in the literature by several orders of magnitude[45].

## Discussion

Cancer cell survival is dependent on specific plasma membrane transport systems that maintain an increased pH$_i$ despite a local acidic environment[1,2,4,7]. NBCn1 plays a key role in breast cancer survival as a base loader that generates a reverse pH gradient[23–25]. Our data provide a structural description of the NBCn1 transport cycle, highlighting the coupling of the conformational states, energetics, and ion transport selectivity. Although we cannot definitively rule out a Na$^+$-HCO$_3^-$ transport mode, our results are most compatible with NBCn1 mediating the highly efficient transport of two Na$^+$ and a CO$_3^{2-}$ ion during an elevator-type transport cycle with a TOR value of ~ 15,000 s$^{-1}$. In this regard, NBCn1 resembles the Na$^+$-coupled SLC4 transporters NBCe1 and NDCBE which are also thought to transport CO$_3^{2-}$ rather than HCO$_3^-$ [39,46,47].

In vivo, NBCn1 is a base loader that transports substrate ions from the extracellular to the intracellular space (OF to IF substrate loaded conformational transition). Our data demonstrate that the NBCn1 conformational changes during its transport cycle are associated with minimal structural reorganization and small energy barriers that could contribute to its very high TOR value. The NBCn1 alternating access transport mechanism can be described as an elevator-type motion involving a small ~5 Å vertical translation of the core domain with the S1 ion binding site with respect to the gate domain (Fig. 3c, Supplementary Fig. S5, Supplementary Movie 1). This vertical motion is most pronounced for the internal core helices (TMs 1, 3, 8, and 10, which are closest to the gate domain; 5.0 Å), whereas the peripheral helices (TMs 2, 4, 9, 11 at the protein/lipid interface) undergo a smaller 2.6 Å displacement. This is expected to induce small reorganizations of the lipid bilayer around the protein with a low energetic penalty. During the NBCn1 vertical elevator-type motion, hydrophobic residues from the helices of the core and gate domains come in contact depending on the position of the core with respect to the gate and form hydrophobic contacts, which impede water permeation as shown in the occluded apo Int conformational state (Supplementary Figs. S7, S8).

The computational modeling data characterizes the steps involved in NBCn1 mediated 2Na$^+$-CO$_3^{2-}$ symport from the extracellular to the intracellular compartment (base loading) that can take place given the ion gradients in breast cancer cells (Fig. 6). In the OF conformational state, where the two Na$^+$ and a CO$_3^{2-}$ ion traverse the OF permeation cavity along their respective permeation pathways (Fig. 3a, b), towards the center of the transporter, the substrate ions bind transiently to the

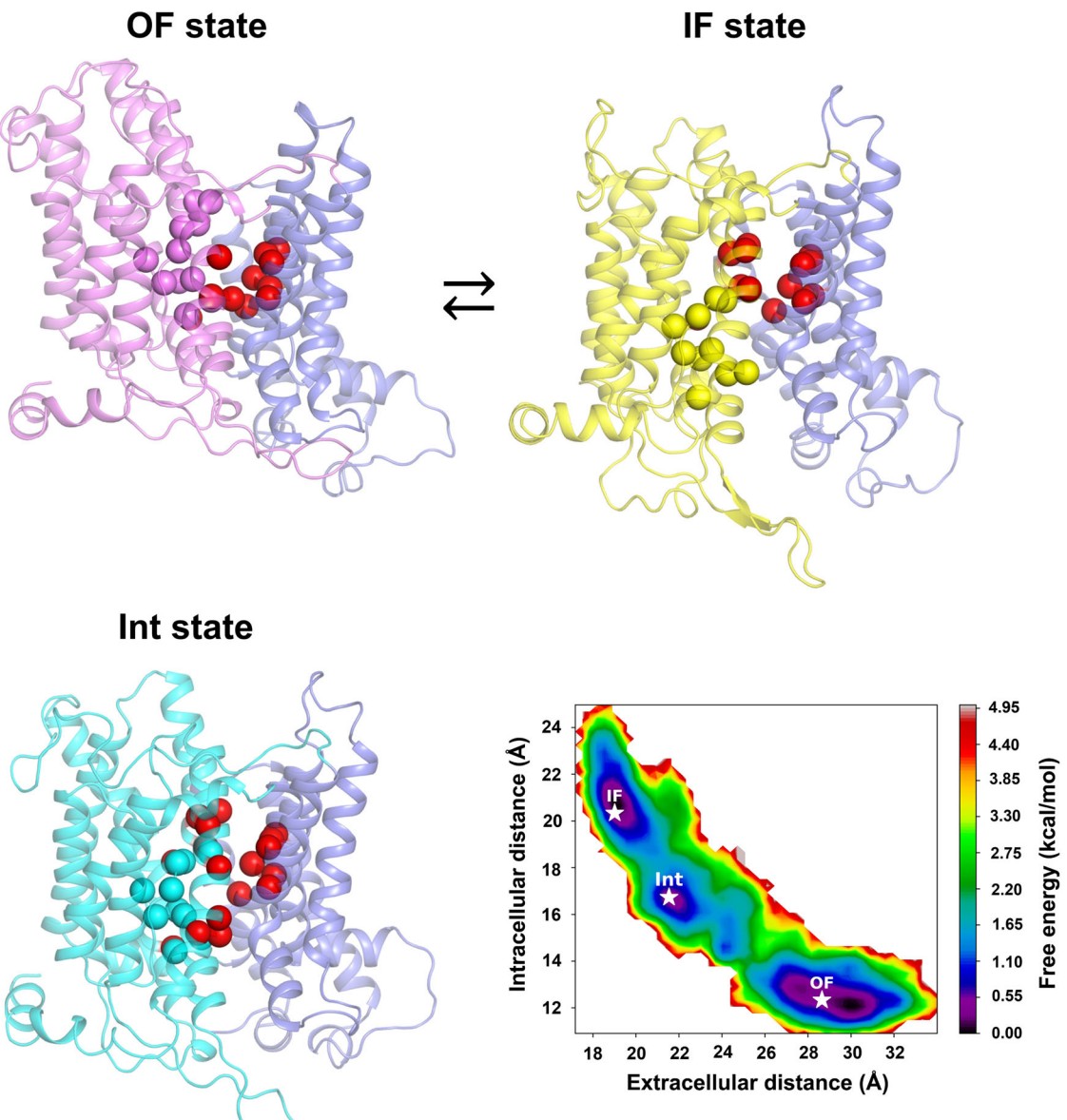

**Fig. 4 | NBCn1 Int state.** Comparison among the OF, IF, and Int states. Gate domain (purple helices), OF core (pink helices), IF core (yellow helices), Int core (cyan helices). The Cα atoms of the ion-binding residues of the core (Fig. 3d) in the OF, IF, and Int states are shown as spheres in the respective core color. The Cα atoms of the hydrophobic gate and core domain residues that come in contact with one another during the OF ↔ IF transition are shown as red spheres. A free energy map obtained from binning of 32 μs of MD trajectories of OF, IF, and various Int conformations generated from Climber[43] is also shown. Clustering done in the space of the intracellular distance (measured between the centers of mass of TM10 and the gate) and extracellular distance (measured between the centers of mass of TM3 and the gate) provides representative OF, IF, and Int states (indicated with a star on the map).

S2$^{MD}$(OF) site (Loaded-S2$^{MD}$(OF); Fig. 6) that consists of residues in proximity or part of S1$^{MD}$(OF) site. Binding of two Na$^+$ and a CO$_3^{2-}$ ion in this area is short-lived and controlled by the negatively charged residues of TM5 and the positively charged residues of TM3 and TM13. The ions then overcome a small energy barrier of ~1.5 kcal·mol$^{-1}$ to move into the S1$^{MD}$(OF) site (Loaded-S1$^{MD}$(OF); Fig. 6), triggering the OF to IF conformational transition. The OF to IF transition proceeds through a hypothesized ion loaded occluded state (Loaded-Int; Fig. 6), in which the core domain assumes an intermediate position between the OF and IF conformational states and the binding pocket with the ions bound at the S1$^{MD}$ site is occluded from the extracellular and intracellular solutions through contact interfaces featuring hydrophobic residues of the gate (TMs 5 and 12) and core (TMs 3, 8, and 10) domains. The S1$^{MD}$(OF) site is positioned at the crosspoint of TMs 3 and 10 and moves

downwards vertically with these TMs during the elevator-type motion of the core domain (Fig. 3, Supplementary Fig. S5, Supplementary Movie 1). Minor local reorganization at the S1 site occurs during its vertical movement in the OF to IF transition, including loss of contact with gate residues Phe727, Ile731 and Lys1100 and enhanced contact with the core domain residues Gly666-Pro670 in TM3 (Fig. 3d), leading to the conversion of site S1$^{MD}$(OF) into a putative S1(Int) site and, afterwards, into the S1$^{MD}$(IF) site (Fig. 6). Once the IF state has been achieved (Loaded-S1$^{MD}$(IF); Fig. 6), the CO$_3^{2-}$ ion dissociates from the S1$^{MD}$(IF) site attracted by the positively charged residues at the dimeric interface lining the anionic pathway toward site S2$^{MD}$(IF) (Loaded-S2$^{MD}$(IF); Fig. 3) and the exit from the IF cavity. The anion pathway between site S2$^{MD}$(IF) and the cavity exit features low energetic barriers (<2 kcal·mol$^{-1}$) and allows swift and unobstructed CO$_3^{2-}$ movement toward the intracellular solution.

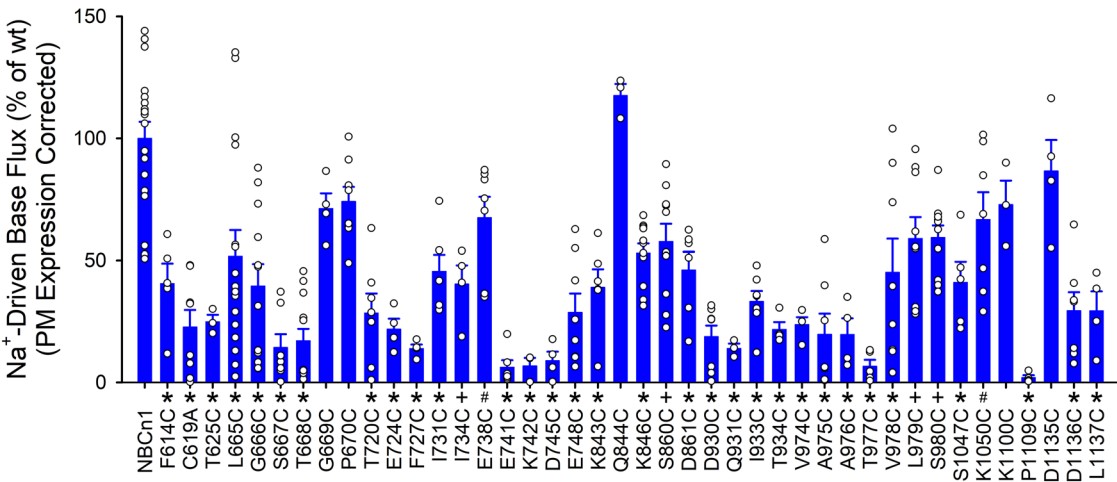

**Fig. 5 | NBCn1 Na⁺-driven base transport function corrected for cell-surface plasma membrane (PM) expression.** The data is depicted as percent of wt-NBCn1 function divided by percent of wt-NBCn1 cell-surface PM expression. NBCn1 wt ($n = 18$ biologically independent experiments) and single cysteine functional mutant data: F614C ($n = 5$, $p < 0.0001$); C619A ($n = 8$, $p < 0.0001$); T625C ($n = 3$, $p < 0.0001$); L665C ($n = 16$, $p < 0.0001$); G666C ($n = 12$, $p < 0.0001$); S667C ($n = 7$, $p < 0.0001$); T668C ($n = 12$, $p < 0.0001$); G669C ($n = 4$, $p = 0.4859$); P670C ($n = 8$, $p = 0.2164$); T720C ($n = 7$, $p < 0.0001$); E724C ($n = 4$, $p < 0.0001$); F727C ($n = 4$, $p < 0.0001$); I731C ($n = 6$, $p < 0.0001$); I734C ($n = 4$, $p = 0.0001$); E738C ($n = 7$, $p = 0.0487$); E741C ($n = 6$, $p < 0.0001$); K742C ($n = 3$, $p < 0.0001$); D745C ($n = 4$, $p < 0.0001$); E748C ($n = 8$, $p < 0.0001$); K843C ($n = 6$, $p < 0.0001$); Q844C ($n = 3$, $p = 0.9992$); K846C ($n = 11$, $p < 0.0001$); S860C ($n = 10$, $p = 0.0001$); D861C ($n = 6$, $p < 0.0001$); D930C ($n = 8$, $p < 0.0001$); Q931C ($n = 3$, $p < 0.0001$); I933C ($n = 7$, $p < 0.0001$); T934C ($n = 4$, $p < 0.0001$); V974C ($n = 4$, $p < 0.0001$); A975C ($n = 7$, $p < 0.0001$); A976C ($n = 4$, $p < 0.0001$); T977C ($n = 5$, $p < 0.0001$); V978C ($n = 8$, $p < 0.0001$); L979C ($n = 9$, $p = 0.0005$); S980C ($n = 10$, $p = 0.0003$); S1047C ($n = 5$, $p < 0.0001$); K1050C ($n = 7$, $p = 0.0379$); K1100C ($n = 3$, $p = 0.8068$); P1109C ($n = 4$, $p < 0.0001$); D1135C ($n = 4$, $p = 1.0000$); D1136C ($n = 7$, $p < 0.0001$); and L1137C ($n = 4$, $p < 0.0001$). One-way ANOVA and Dunnett's test were used to compare multiple study group means with wt-NBCn1. Statistically significant results differing from wt-NBCn1 are depicted as mean ± SEM (#$p < 0.05$, +$p < 0.01$ and *$p < 0.0001$). Source data are provided as a Source Data file. Residues are numbered based on NBCn1-A for structural comparison.

Na⁺ ions exit site S1$^{MD}$(IF) and the IF cavity either together with $CO_3^{2-}$ toward the S2$^{MD}$(OF) site or along the cation permeation pathway that includes Thr720, Glu724, Asp1135, Asp1136 and Leu1037 until an apo-IF state is achieved (Fig. 6). The cycle is concluded by an apo-IF to apo-OF transition through an apo-Int state with hydrated binding pocket and S1(Int) site (Figs. 4, 6, and Supplementary Fig. S7).

The finding that NBCn1 appears to preferentially utilize an electroneutral $2Na^+$-$CO_3^{2-}$ transport mode ensures that despite the presence of a reverse pH gradient, the inward Na⁺ thermodynamic driving force significantly exceeds the outward $CO_3^{2-}$ driving force ensuring continued NBCn1-mediated base influx over a wide range of extracellular pH values and transcellular pH gradients (Supplementary Figs. S12, S13). In this way, NBCn1 is uniquely positioned to mediate base influx unlike other SLC4 transporters whose transport modes and associated thermodynamic driving forces don't support base loading at the intracellular and extracellular ion chemistries typical of breast cancer cells[3,15,31,48–50].

An important contribution to the energy demand and glycolytic flux of breast cancer cells is the need for ATP to drive the plasma membrane Na⁺/K⁺-ATPase[51]. The high NBCn1 TOR value of ~15,000 s⁻¹ (Supplementary Table 3), provides breast cancer cells with an additional survival advantage in that mediating a given rate of cellular base influx theoretically would require fewer plasma membrane NBCn1 transporters, thereby lowering the energy expenditure utilized in the processes of transporter synthesis and degradation. Furthermore, by transporting $CO_3^{2-}$ rather than $HCO_3^-$ in each transport cycle, the ensuing extracellular and intracellular pH changes are predictably greater[46,47].

## Methods

### NBCn1 purification
N-terminally Strep(II)-tagged wt-human NBCn1-A (NP_003606.3, 1214 aa) in pcDNA3.1(+) was transfected into HEK293 cells (ATCC) using polyethyleneimine. The cells were grown on 10-cm plates for ~24 h in Dulbecco's Modified Eagle's medium (Thermo Fisher Scientific) and 5%

fetal bovine serum (Thermo Fisher Scientific). Cells were collected from ~200 plates, washed in PBS, and pelleted at 2,000 g for 10 min. Then cells were solubilized for 30 min in 2% Triton X-100 (Sigma-Aldrich) in buffer A containing 50 mM Tris-HCl (Sigma-Aldrich), 500 mM NaCl (Sigma-Aldrich), pH 7.5, and complete protease inhibitor cocktail (Bimake). Following centrifugation at 20,000 x $g$, the supernatant was loaded onto the 5-ml StrepTrap HP column (GE Healthcare). The StrepTrap HP column was rinsed with buffer A containing 0.01% lauryl maltose neopentyl glycol (LMNG, Anatrace), and bound protein was eluted with buffer A containing 2.5 mM D-desthiobiotin (Sigma-Aldrich). A Superose 6 size exclusion column (GE Healthcare) was used to further purify NBCn1 with 20 mM Tris-HCl, 115 mM NaCl, 25 mM NaHCO₃, pH 7.4, and 0.01% LMNG. The peak corresponding to dimeric NBCn1 (Supplementary Fig. S1) was collected and further concentrated on Ultracel 100 kDa MWCO (Amicon).

### CryoEM sample preparation and data acquisition
Vitrobot Mark IV (Thermo Fisher) was set to 4 °C with 100% humidity. Filter papers were loaded an hour earlier before the cryo-grid preparation. The cryoEM grids were prepared by applying 3.5 μl of NBCn1 sample (~1.5 mg·ml⁻¹) to a freshly glow-discharged Nickel-Titanium M01-Au300 mesh R1.2/1.3 grid (ANTcryo) and blotted with filter paper to remove excess sample for 2 s followed by a rapid plunge-freezing into liquid ethane. The grids were stored in liquid nitrogen until loaded into a Titan Krios electron microscope for data acquisition, operated at 300 kV. Super-resolution movie stacks were recorded by a Gatan Gif Quantum K3 Summit direct electron detector (Gatan) in EFTEM mode with 20 eV slit at a nominal magnification of 81,000x, resulting in a calibrated pixel size of 0.55 Å. SerialEM software was used for automated data collection. Based on the calibrated dose rate 28.6 e⁻·pix⁻¹·s⁻¹, the exposure time for each movie stack was set to 2 s with a 0.05 s frame rate, yielding 40-frame stacks with a total electron dose of ~47 e⁻·(Å²)⁻¹. The defocus range was set from −1.8 μm to −2.4 μm. During data collection, cryoSPARC[37] was used to perform patch

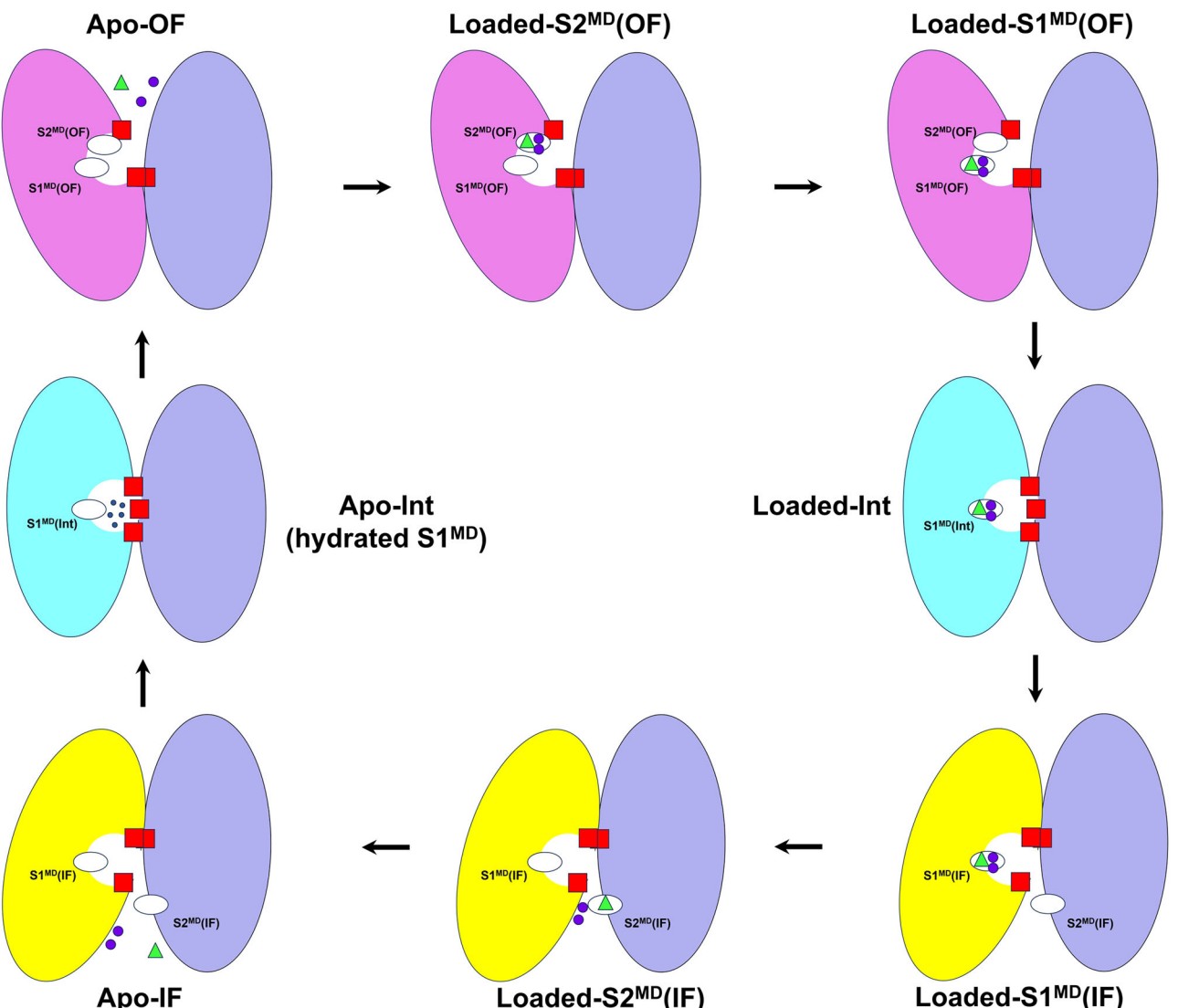

**Fig. 6 | Schematic overview of the NBCn1 OF ↔ IF elevator-type transport cycle.** Gate domain (purple oval), core domain (pink, cyan, and yellow oval in the OF, Int, and IF states, respectively), gate and core domain hydrophobic residues from the contact interface (red squares), ion binding sites (white ovals), $CO_3^{2-}$ ions (green triangles), $Na^+$ ions (purple spheres), water molecules (small dark blue spheres).

motion correction and patch CTF estimation for each generated stack in real time. Dosed-weighted micrographs were binned to a pixel size of 1.1 Å·pix$^{-1}$ for the following data processing. Streamlined blob picker and 2 d classification were performed in cryoSPARC[37] for real-time data quality assessment.

## CryoEM data processing

A total number of 16,274 micrographs were selected through manually curated exposures in cryoSPARC[37] (Supplementary Fig. S1a), and good 2D average classes generated during data collection were used for Topaz training and Topaz picking[52]. After particle picking and particle extraction, 2,560,294 particles were sent to two rounds of 2D classification. The 2D classes with clear secondary structure features were selected in each round and provided 1,240,050 good particles for the following 3D classification and refinement. An initial 3D reference was generated from the selected particles via Ab initio reconstruction and was sent to hetero refinement with four classes in cryoSPARC[37]. One class with good secondary structure features was selected and 666,750 particles were sent to another round of 2D classification. 558,428 particles were selected and reextracted with a larger box size of 384

pixels. The reextracted particles were sent to non-uniform (NU) refinement generating a reconstruction at 3.6 Å[53].

The particles were imported into RELION[36] to perform further 3D classification with a mask focusing on TMD. After 2 rounds of 3D classification, 3 dimeric states were detected depending on the ability to resolve TM13 and TM14 in each monomer. In state 1 homodimers both monomers had resolved TMs13/14; state 2 hybrid dimers had one monomer with resolved TM13/14; state 3 homodimers lacked resolved TM13/14 in each monomer (Supplementary Fig. S1a). The three different dimeric states were sent to NU refinement in cryoSPARC[37] that generated three density maps at 3.5 Å, 3.4 Å, and 3.3 Å, respectively. Particles belonging to the state 1 and 2 dimer classes were then combined and refined, yielding a final reconstruction at 3.3 Å.

## Model building and refinement

De novo atomic model building was performed automatically by using EMBuilder[54] and manually built in Coot[55] based on the 3.3 Å resolution map. The high quality of the density map allowed us to build a NBCn1 model that consisted of residues 581 to 1207. Amino acid assignment was achieved based on the clearly defined densities for bulky residues

(phenylalanine, tryptophan, tyrosine, arginine and lysine). The model was refined against the 3.3 Å map using phenix.real_space_refine[56] with the secondary structure restraints and non-crystallography symmetry applied. To evaluate the signal of ions difference map, the experimental EM density map minus the simulated map from the refined model without ions was calculated. The distances between ions and coordinating residues were adjusted according to Harding[57]. The statistics for the model geometries were generated using Molprobity[58] (Supplementary Table 1) and EMRinger[59]. Model and density figures were prepared in UCSF Chimera[60] or Pymol[61].

## Homology Modeling of the NBCn1 IF conformation

A homology model of the IF state of NBCn1-A was built with the Swiss-Model server[41], using the recently resolved IF cryoEM structure of the SLC4 Cl⁻/HCO₃⁻ exchanger, AE2 (PDB ID 8GV9)[40]. The long EL3 in the IF state was modeled based on the one resolved in the OF cryoEM structure of NBCn1. Validation of the IF homology model of NBCn1 was done by close inspection of the amino-acid position and orientation of key transport regions in the protein matrix and cross-validation with previous functional and structural data for various members of the SLC4 family: AE1[62], AE2[40], NBCe1[38], NDCBE[39], SLC4A11[63], and with NBCn1 functional mutagenesis data obtained in this study.

## OF and IF conformation SILCS calculations

The SILCS simulations were performed with single monomers of NBCn1 in the OF (cryoEM structure) and IF conformation (homology model) with truncated EL3 between residues F752 and P822. SILCS[42] makes use of the GROMACS v.2022.6 MD simulation package[64] and combines Grand Canonical Monte Carlo/Molecular Dynamics (GCMC/MD) methodology in oscillating excess chemical potential simulations[65] to yield fragment maps for several small organic probes (hydrogen bond donors, hydrogen bond acceptors, aromatic, aliphatic, and charged functional groups) with different physical and chemical properties in and around the protein. Explicit water molecules (concentration 55 M) and the aforementioned probes (concentration 250 mM for each probe) compete for coordination to different parts of the protein matrix to generate the fragment maps, which pinpoint protein areas where specific fragments tend to accumulate and coordinate. Briefly, the single protein monomer structure was embedded in a 1-palmitoyl-2-oleoylphosphatidylcholine (POPC) and cholesterol membrane in a 9:1 ratio, respectively, using the SILCS Software v.2023.1.1. The system size was set to 120 × 120 Å in the x and y axis, with the protein center of mass placed in the center of the bilayer. The SILCS protocol generates ten different systems, where the protein embedded in a bilayer is solvated with explicit water molecules and with the eight types of probe molecules, randomly placed in each of the ten independent replicas. Protein and lipids are represented by SILCS using the CHARMM36m[66] and CHARMM36[67] force fields, respectively, and water molecules with the CHARMM TIP3P model[68]. The parameters for the probe molecules were gathered from the CHARMM General ForceField (CgenFF)[69]. The SILCS simulations were performed with a time step of 2 fs, and temperature and pressure were kept at 298 K and 1 bar using the Nose-Hoover Thermostat[70] and the Parrinello-Rahman Barostat[71], respectively. The minimization and equilibration steps were used as specified by the default parameters described in Ustach et al[42]. During the production runs, restraints were placed on the alpha-carbon atoms of the protein, with a force constant of 0.1 kcal·mol⁻¹·Å⁻². In total, 100 cycles of GCMC/MD comprising 200,000 GCMC steps and 1 ns of MD simulation were performed per replica, yielding a total of 1 μs MD data for the NBCn1 OF and IF conformations. From the average of the simulation data from each of the 10 independent simulations, solute atoms were binned onto a grid of 1 Å voxel followed by a Boltzmann-based transformation to yield the Grid Free Energy (GFE maps) representing the affinity of the probe molecules in and around the protein[65].

## MD simulations with position restraints on the bound anion in OF and IF NBCn1 with 2Na⁺-CO₃²⁻ ion load

For assessment of permeation energetics and identification of putative binding sites in the OF and IF cavities of NBCn1, truncated models of OF and IF NBCn1 missing the EL3 between residues Phe752 and Pro822 and loaded with 2Na⁺-CO₃²⁻ were subjected to MD simulations, in which the C atom of the $CO_3^{2-}$ ion was restrained to motion only within the xy-plane by imposing a force constant of 1000 kJ·mol⁻¹·nm⁻² (239 kcal·mol⁻¹·nm⁻²) on its motion along the z-axis. For these simulations, the truncated OF and IF NBCn1 structures were embedded in a POPC lipid bilayer (109 POPC molecules in upper leaflet, 102 POPC molecule in lower leaflet) in a periodic box of size 96x96x108 Å, with 22.5 Å water layer at the extra- and intra-cellular side of the membrane and 150 mM NaCl solution using CHARMM-GUI[72] 71 windows per state were then prepared as independent simulations by moving the 2Na⁺-$CO_3^{2-}$ ion load by 0.3 Å upwards or downwards along the z-axis within the OF or IF cavity, respectively, starting from the initial position of the ions in S1$^{cryoEM}$(OF) and its equivalent in the IF state. The two Na⁺ ions were unrestrained in all directions. The simulations were performed with GROMACS v.2022.6[64], with the CHARMM36m[66] (proteins), CHARMM36[67] (lipid, ions), and CGenFF[69] ($CO_3^{2-}$) parameters and the TIP3P water model[68]. A brief minimization (5000 steps with steepest descent algorithm) and the standard 6-step CHARMM-GUI equilibration protocol with gradual relaxation of the position restraints on lipid, protein, and solvent atoms were performed before the production runs. The 100 ns production runs per window were performed with the following set-up: 2 fs time-step, 310.15 K temperature maintained with Nose-Hoover thermostat[70] (t$_t$ 1.0 ps), 1 atm semi-isotropic pressure maintained with C-rescale barostat[73] (compressibility $4.5 \times 10^{-5}$ bar⁻¹, t$_p$ 5.0 ps), LINCS algorithm for bonds involving H-atoms[74], Verlet cutoff scheme with r$_{list}$ 1.2 nm[75], smooth Particle Mesh Ewald electrostatics with r$_{coulomb}$ 1.2 nm cut-off scheme[76] for the VdW interactions with r$_{vdw}$ 1.2 nm and r$_{vdw}$ switch 1.0 nm. Analysis of the trajectories was done with the VolMap tool of VMD 1.9.3[77] and the density-based free energies were plotted using the plot_free_energy tool as implemented in PyEmma v.2.5.12[78] using 50 bins in each dimension.

## Unrestrained 200 ns MD simulations of OF and IF NBCn1 with a 2Na⁺-CO₃² ion load and 1 μs MD simulations of OF and IF apo-NBCn1

200 ns long MD simulations of NBCn1 loaded with a 2Na⁺-CO₃²⁻ ion load were performed with monomers of the OF conformation resolved by cryoEM and the IF homology model of the TMD, in three independent replicas. The initial placement of the ions was guided by the cryoEM densities resolved in S1$^{cryoEM}$(OF) site reported here, the ion coordination sites resolved in the IF conformation of AE2[40] and SLC4A11[63] and the cationic and anionic SILCS maps. The prepared ion-loaded IF and OF systems were embedded in a POPC bilayer (186 and 180 POPC molecules in the upper and lower leaflet, respectively), in a rectangular ~120 x 120 x 140 Å periodic box and solvated by a ~20 Å layer of 75 mM NaCl + 37.5 mM Na₂CO₃ + 37.5 mM NaHCO₃ simulation solution with the CHARMM-GUI[72] server. A 6-step equilibration procedure, with step-wise decrease of restraints placed on the protein backbone atoms was then performed, followed by 200 ns unrestrained production runs with the following settings: force fields CHARMM36m[66], CHARMM36[67], CGenFF[69] (see above); and TIP3P model[68] for water molecules; semi-iso-thermal-isobaric (NPaT) conditions at 310.15 K and 1 atm, controlled via Langevin dynamics with damping coefficient 1.0 ps⁻¹ long-range electrostatic interactions evaluated with Particle Mesh Ewald[76]; and non-bonded interactions cut-off and switch off values set at 12 and 10 Å, respectively. The calculations were performed with NAMD 2.14[79]. Ion time series from these simulations are presented in Supplementary Fig. S6.

In addition, three 1 μs long independent MD simulations per conformation were performed for monomers of the IF and OF

apo-NBCn1 after truncation of EL3. The systems were prepared as described above and after 10 ns unconstrained MD simulations with NAMD 2.14[79] were submitted to 1 μs MD simulations with ANTON2 using the following settings: ANTON2 software version 1.59.0c7 (https://www.psc.edu/resources/anton-2/), 310.15 K temperature controlled by a Nose-Hoover thermostat[70], 1 atm pressure controlled by a MTK barostat[80], multigrator scheme for evolution of the MD trajectories with RESPA algorithm[81] for evaluation of electrostatic and non-bonded interactions. RMSD plots for the six 1 μs MD simulations are presented in Supplementary Fig. S14 and demonstrate structural stability of the OF and IF NBCn1 models and convergence of the MD trajectories. In replica 3 of the IF NBCn1 state, a concerted upward motion of TMs 1 and 3 with respect to the gate leads to a slight increase in RMSD indicating initial stages of formation of a putative intermediate state. However, no occlusion and interruption in the water and ion dynamics was observed in the IF cavity at the end of the 1 μs MD simulation in replica 3.

Generation of averaged ion density maps was done with the VolMap and VolUtil tools of VMD 1.9.3[77]. In-house TCL and CHARMM scripts were used for contact frequency and ion time series analysis, respectively, for further identification of ion coordination sites. The first 10 ns of the simulated trajectories were discarded for these analyses.

### Absolute free energy of binding calculations

Absolute binding free energies for the $2Na^+-CO_3^{2-}$ or $Na^+-HCO_3^-$ ion loads in the $S1^{MD}$(OF) and $S1^{MD}$(IF) sites were done with truncated OF and IF models of NBCn1 lacking the EL3 between residues Phe752 and Pro822 extracted from the 200 ns MD simulations (see above). The input files were prepared with the Absolute Free Energy Binder tool of CHARMM-GUI[72] and featured NBCn1 monomers, either in the OF or IF state, embedded in a POPC bilayer (109 POPC molecules in upper leaflet, 102 POPC molecule in lower leaflet) in a rectangular periodic box of size 96x96x103 Å, with 20 Å water layer at the extra- and intra-cellular side of the membrane and 150 mM NaCl solution. Hydration calculations were performed for $CO_3^{2-}$ and $HCO_3^-$ in a cubical water box of the size 53 x 53 x 53 Å (a total of 4715 water molecules) in the presence of 2 or 1 $Na^+$ ions. The free energy calculations were done with NAMD 2.14[79] and the CHARMM36m[66], CHARMM36[67], and CGenFF[69] parameters (see above); and the TIP3P water model[68] using the λ-Replica-Exchange Molecular Dynamics protocol[82] to enhance sampling. To that end, the free energy calculations were first subjected to the standard CHARMM-GUI six-wise step procedure of minimization and equilibration and then concurrently done in 32 linearly-spaced replica windows with λ values from 0 to 1, with the Metropolis–Hastings exchange criterion[83] for exchanges between different windows. Exchanges between different replicas were observed in all replicas during the REMD simulations. Postprocessing was done with the provided CHARMM-GUI overlap-sampling estimator scripts[82]. The MD protocol followed that of the NAMD[79] equilibration of production runs (see above). All FEP/λ REMD simulations were run for 10 ns production runs per window. Convergence was achieved in all calculations after 5 ns. The last 4 ns of sampling were used for calculation of the means and standard deviations of the free energy of binding values reported in Supplementary Table 2.

### Modeling of the OF ↔ IF transition

To map the transition from the outward-facing (OF) to the inward-facing (IF) state of the protein, we employed the Climber method[43], a specialized non-linear morphing technique. Climber works by progressively aligning the interatomic distances of the protein in the OF state with those in the IF state by introducing harmonic restraints into the system's energy function, which penalizes the differences between corresponding distances. By iteratively adjusting the force constants

on these restraints, the method minimizes these differences, effectively "pulling" the protein structure along a transition pathway. Compared to linear interpolation techniques, Climber allows for more realistic and less forceful transitions, preserving critical structural features. Using the Climber method, 105 distinct conformations that trace the transition from the OF to the IF state were generated using truncated models of the OF and IF state, which lacked the EL3 residues between Ala751 and Asp823. Afterwards, 16 representative structures from these 105 conformations, that span the conformational space between the two states, were selected and submitted to unrestrained equilibrium MD simulations. To that end, the residues in the 16 structures were renamed to switch from the ENCAD forcefield[84] used by Climber, to CHARMM36 force field[67]. The GROMACS pdb2gmx tool was used for generation of the topology files for the protein molecule and for addition of the missing hydrogen atoms (via the -ignh option). The protein structure was minimized for 5000 steps or until a maximum force in the system below 1000 kJ mol$^{-1}$ nm$^{-1}$ (239 kcal·mol$^{-1}$·nm$^{-1}$) was reached using GROMACS v.2021.4[64] with the CHARMM36m[66] forcefield to remove any abnormal contacts that could arise while placing the hydrogens. The proteins were then embedded into a POPC lipid bilayer (233 POPC molecules in total) in a box of size 100 ×100 x 108 Å with 20 Å layer of water at the intra- and extra-cellular side of the membrane and 75 mM NaCl + 37.5 mM $Na_2CO_3$ + 37.5 mM NaHCO$_3$ solution using CHARMM44b1 scripts generated by CHARMM-GUI[72]. For consistency across the simulations, the number of water molecules was adjusted to the same number (20,206 water molecules) in all 16 windows of the OF ↔ IF transition. The system was equilibrated in six steps over 100 ns following the CHARMM-GUI scheme of gradual reduction of restraints on the protein backbone, side chains, lipid molecules, and dihedral angles to ensure structural stability. Following equilibration, two replicas of equilibrium MD simulations (2 × 16 windows), each running independently for 1 μs per window, for a total of 32 μs of unrestrained sampling were performed at a constant temperature of 310.15 K, maintained by the v-rescale thermostat in GROMACS v.2021.4[85], with a time constant of 1 ps. Pressure was kept at 1 bar with a compressibility of $4.5 \times 10^{-5}$ bar$^{-1}$ and a time constant of 5 ps, using the c-rescale barostat[73] for equilibration and the Parrinello-Rahman barostat[71] for production runs. The LINCS algorithm[74] was employed to constrain all bond lengths involving H-atoms, ensuring accurate representation of the protein and lipid structures throughout the simulations. The Verlet cut-off scheme as implemented in GROMACS v.2021.4[75] was used with a cutoff radius of 12 Å. In addition, a force-switch modifier was used with a cutoff radius of 10 Å. Electrostatics were modeled using the Particle Mesh Ewald (PME)[76] with a short-range cutoff radius of 12 Å. Analysis of the produced trajectories was done with Pyemma v. 2.5.12[78]. The center of mass distances of the 8 transmembrane helices of the core domain from the gate domain were used as collective variables. This simulation data was discretized into 1200 clusters using the Kmeans clustering algorithm (as implemented in Pyemma). The clusters close to the stationary states were used to extract structures corresponding to the IF, OF, and an intermediate state. These structures were then used to compute water maps with the VolMap tool of VMD 1.9.3[77].

Additional details for all simulations reported here can be found in Supplementary Table 4.

### NBCn1 transport assays

The functional NBCn1 transport assays were performed using a modified custom-designed dual excitation microscope-fluorometer to measure pH$_i$ changes in HEK293 cells with BCECF ratiometrically[44,86]. HEK293 cells growing on 25 mm PEI-coated glass round cover slips were transiently transfected with human wt-NBCn1, the various NBCn1 mutant constructs (primers (Integrated DNA Technologies) used for mutagenesis are shown in Supplementary Data 1 file) or a pcDNA3.1(+) empty expression vector (used for background subtraction). For these

studies the human NBCn1-H variant (NP_001308036.1, 1095 aa) in pcDNA3.1(+) was used because of its increased transport function. The assays were performed 24 h following transfection with Lipofectamine 2000. The cells were initially loaded in the following Na⁺-free, Cl⁻-containing, HCO₃-free solution: 140 mM tetramethylammonium chloride, 2.5 mM K₂HPO4, 1 mM CaCl₂, 1 mM MgCl₂, 5 mM HEPES, 5 mM dextrose, pH 7.4, and 30 µM EIPA. After washout with the same solution and a steady baseline, the cells were then bathed in a Na⁺-free, Cl⁻- and HCO₃⁻-containing solution: 115 mM TMA-Cl, 2.5 mM K₂HPO₄, 1 mM CaCl₂, 1 mM MgCl₂, 24 mM TMA-HCO₃⁻, 5 mM dextrose, 5% CO₂, pH 7.4, and 30 µM EIPA. Following a steady state, NBCn1 transport was initiated by switching to the following Na⁺-, Cl⁻-, and HCO₃⁻-containing solution: 115 mM NaCl, 2.5 mM K₂HPO₄, 1 mM CaCl₂, 1 mM MgCl₂, 24 mM NaHCO₃, 5 mM dextrose, 5% CO₂, pH 7.4, and 30 µM EIPA. $CO_3^{2-}$ flux was calculated by measuring the initial rate (10–15 s) of change of $[H^+_{in}]$ $(d[H^+_{in}] \, dt^{-1})$ after the Na⁺ addition and multiplying by the total cell buffer capacity (intrinsic ($\beta i$) plus bicarbonate ($\beta HCO_3$)).

### Plasma membrane NBCn1 protein labeling

A sulfo-NHS-SS-biotin labeling kit (Thermo Fisher Scientific) was used to label and pull down wild-type and mutant NBCn1 plasma membrane proteins according to the following protocol. The transfected cells were washed with PBS (pH 8.0) and then incubated with 1.1 mM sulfo-NHS-SS-biotin (Thermo Fisher Scientific) in PBS (pH 8.0) for 30 min after which the reaction was stopped with 50 mM Tris-HCl buffer containing 140 mM NaCl, pH 8.0. The cells were then washed and collected with PBS (pH 8.0), and lysed on ice in 150 mM NaCl, 1% (vol/vol) Igepal (Sigma-Aldrich), 0.5% sodium deoxycholate (Thermo Fisher Scientific), 5 mM EDTA (Sigma-Aldrich), and 10 mM Tris-HCl, pH 7.5, with protease inhibitors (Roche Life Sciences). Following 10 min centrifugation at 16,000 x $g$ (4 °C), the plasma membrane and lysate were collected and incubated with 50 µl of streptavidin-agarose resin (Thermo Fisher Scientific) on a rotating shaker at 4 °C for 4 h. The resin was pelleted following brief centrifugation and washing with the lysis buffer, bound proteins were eluted with 2 × SDS buffer containing 2% β-mercaptoethanol (EMD Millipore, Billerica, MA) for 5 min at 60 °C. Plasma membrane and lysate protein samples were initially resolved on 7.5% polyacrylamide gels and then transferred to polyvinylidene difluoride (PVDF) membranes (GE HealthCare). Protein expression levels were assessed by probing the blots with the NBCn1 antibody (Anti-SLC4A7/NBCn1antibody (ab82335, Abcam), 1:10,000 dilution) in TBSTM buffer (137 mM NaCl, 20 mM Tris-HCl, pH 7.5, 0.1% (vol/vol) Tween 20 and 5% (w/vol) nonfat milk) for 1 h at room temperature. The blots were washed with TBST and probed with a horseradish peroxidase (HRP)-conjugated AffiniPure Mouse Anti-Rabbit IgG (H + L) secondary antibody (Jackson ImmunoResearch Laboratories, Inc.) at 1:10,000 dilution in TBSTM buffer (room temperature; 1 h). Following washing of the blots with TBST, ECL Western Blotting Detection Reagent (GE Healthcare) was used for detection and the band intensities were quantified.

### NBCn1 ion turnover rate (TOR)

An NBCn1-H construct containing a V5-epitope (underlined) inserted into NBCn1-H-EL3 (656-SNET<u>GKPIPNPLLGLDST</u>LAQW-677) was used in these studies. A) Transport assays: The functional transport experiments were performed as described above. B) Plasma membrane NBCn1 transporter number per cell: The number of NBCn1 plasma membrane transporters was determined by biotinylation and pulling down of plasma membrane proteins as described above. A standard curve was generated using known amounts of a peptide that contained the V5 epitope tag[44,45] (Multiple tags (GTX130343-pro, GeneTex peptide). Various dilutions of the peptide were used to obtain standards from 0.0625 ng (1.36 fmol) - 1 ng (21.7 fmol) that were loaded along with biotinylated V5-tagged NBCn1 protein at the same time onto the same 4–15% polyacrylamide gels. Following transfer to polyvinylidene

difluoride (PVDF) membranes (GE HealthCare) and probing with a V5 monoclonal antibody (R960-25, Thermo Fisher Scientific; dilution 1:10,000) in TBSTM buffer (137 mM NaCl, 0.1% (vol/vol) Tween 20 and 5% nonfat milk (w/vol)) for 1 h at room temperature, the blots were washed with TBST and labeled for 1 h at room temperature with secondary antibody (Peroxidase AffiniPure Donkey Anti-Mouse IgG (H + L) antibody; Jackson ImmunoResearch; 1:10,000 dilution). After rinsing the blots with TBST the bands detected using ECL Western Blotting Detection Reagent (GE HealthCare). The complete set of V5 standards and plasma membrane NBCn1 protein was detected at the same time on the same x-ray film at the same exposure. To determine the total number of plasma membrane NBCn1 transporting monomers, we measured and combined the intensity of the monomeric and dimeric transporter bands in comparison to the intensities of the V5 standard curve bands. C) TOR calculation: NBCn1 flux (mM·s⁻¹) was converted to mol·s⁻¹·cell⁻¹ by multiplying by the measured cell volume (Cellometer Auto T4; Nexcelom Bioscience). In the immunoblot experiments, the total number of moles of plasma membrane NBCn1 protein monomers was expressed in units of mol·cell⁻¹ by dividing by the number of cells (Cellometer Auto T4; Nexcelom Bioscience) from which the biotinylated plasma membrane protein samples that were loaded onto the gels were derived. NBCn1 TOR (s⁻¹) was calculated by dividing the transporter flux (mol·s⁻¹·cell⁻¹) by the plasma membrane transporter monomer number (mol·cell⁻¹).

### Statistical analysis

The results are shown as mean ± SEM, and group means are considered significantly different when $p < 0.05$. One-way ANOVA and Dunnett's test were used to compare multiple experimental replicate group means. In the absolute free energy of binding experiments, the temporal fluctuation of the data points in the last 4 ns of each simulation is depicted as mean ± SD.

### Reporting summary

Further information on research design is available in the Nature Portfolio Reporting Summary linked to this article.

## Data availability

The final cryoEM density map of NBCn1 has been deposited to the Electron Microscopy DataBank (EMDB) under the accession code EMD-70906 (the main cryoEM density map). The final atomic model has been deposited into the Protein Data Bank (PDB) under the accession code 9OVR (OF conformation). The data supporting the findings of this study are available in the manuscript, Supplementary Information file, source data file, and Supplementary Data files. The source data underlying Fig. 5 and Supplementary Figs. 1b; 9a, b; and 11b are provided as a Source Data file. Initial and final steps from the various MD trajectories are provided as Supplementary Data files 2 to 5. Climber intermediates are provided as Supplementary Data 6 file. Initial structures for the free energy of binding calculations are provided as Supplementary Data 7 file. Source data are provided with this paper.

## Code availability

The in-house scripts used for trajectory analysis are available at GitHub: [https://github.com/hzhekova/Scripts_for_NatComm_NBCn1].

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

## Acknowledgements

I.K. was supported by funds from the NIH (R01 DK077162), the Smidt Family Foundation, the Kleeman Fund and the Factor Family Foundation. Z.H.Z.'s group was supported by the NIH (R01GM071940 and R01 DK077162). D.P.T.'s group was supported by the NIH (R01 DK077162) and the Canadian Institutes for Health Research (PJT-180245), with further support from the Canada Research Chairs program. Calculations were carried out on resources from the Digital Research Alliance of Canada funded by the Canada Foundation for Innovation and partners. Part of the production simulations were performed on the Anton2 computer provided to I.K. by the Pittsburgh Supercomputing Center (PSC) and DE Shaw Research through Grant R01GM116961 from the NIH. We acknowledge the Electron Imaging Center for NanoSystems for the use of cryoEM resources supported by the University of California, Los

Angeles, the NIH (1S10RR23057), and the NSF (DBI-1338135 and DMR-1548924). The content is solely the responsibility of the authors and does not necessarily represent the official views of the National Institutes of Health.

## Author contributions

I.K. conceived of the study. W.W. collected and analyzed the cryoEM data, and Z.H.Z. supervised these experiments. L.K. and D.A. performed large-scale expression of NBCn1 in HEK293 cells, and K.T. performed purification and biochemical characterization. L.K., R.A, D.A., and N.A. performed functional mutagenesis experiments. H.R.Z., B.S.D., E.G.A., and D.P.T. designed and performed the computational modeling. I.K., H.R.Z., W.W., D.P.T., and A.P. wrote the manuscript. All authors contributed to the critical revision of the manuscript and approved the final version.

## Competing interests

The authors declare no competing interests.
