## [Transparent Peer Review file · Nature Communications]

CryoEM and Computational Modeling Structural Insights into pH Regulator NBCn1

Corresponding Author: Dr Ira Kurtz

Version 0:

Reviewer comments:

Reviewer #1

(Remarks to the Author)

The manuscript submitted by Weiguang Wang et.al describes the first high resolution CryoEM structure of human NBCn1/SLC4A7 protein in complex with putative transporting ligands. NBCn1 is a highly important research target as the protein is crucial for breast cancer cells survival at acidic environment caused by aerobic respiration and lactic acid production. The authors have suggested new transporting ligands of the protein, unlike other close SLC4 members that transport HCO₃⁻, NBCn1 is actually transporting CO₃²⁻ together with 2 Na⁺, adding up to an electroneutral transporter.

This new mechanism is used to explain why NBCn1, not other SLC4 members are so relevant to the dramatic lower extracellular pH of breast tumor cell. The authors have done mutagenesis assay, MD simulation and thermodynamic calculation to show that the transporting ligand is suitable for base loading into the cancer cell at near physiological conditions. Also the authors have measured the turnover rate of the transporter and shown NBCn1 to be one of the fastest symporters reported to date.

Overall the manuscript is well written, the result is novel, and would provide new insight that is relevant to many both in transporter structure community and breast cancer research community. But I do have concerns about the new mechanism as the authors have not discussed about the possibility AT ALL that NBCn1 could still be a Na⁺ and HCO₃⁻ co-transporter. As the authors have shown, the HCO₃⁻ / CO₃²⁻ concentration ratio is 24.3/0.0367 ~ 600 fold at pH7.4 and 6.09/0.00232 ~2600 folds at pH6.8, which means that the transporter have a much higher probability to encounter HCO₃⁻ ion at physiological conditions, how could this happen if NBCn1 is the fastest transporter?

It is not very easy to tell HCO₃⁻ and CO₃²⁻ from CryoEM density as both ligands have the same triangle shape. Further, it is very important to show the putative ion density in the CryoEM density with different contour level as it's usually not to easy to tell sodium ion with water, either. Could there be possibility that one of the proposed ion densities is Na and the other one is water? That make the transporter a water dependent Na⁺ and HCO₃⁻ co-transporter, which is the ligands previously suggested by most of the researcher with less change comparing to the current mechanism suggested by the authors.

I would like the author to consider that possibility and compare with their current analysis why they prefer one from another as I could not see any violation of a water dependent Na⁺ and HCO₃⁻ co-transporter with the reasoning they have made in the manuscript, both MD and thermodynamic calculation.

If the authors could clear my concern, I would be very thankful and suggest the manuscript to be accepted in NC, but currently I do worry that the author have come to a too bold hypothesis with limited information from CryoEM density, which could be easily misinterpreted.

Reviewer #2

(Remarks to the Author)

The manuscript by Wang et al. reports a combined experimental and computational study of the structural dynamics of NBCn1 transporter that helps with the survival of the breast cancer cells. They have determined a cryoEM structure for this transporter in the outward-facing (OF) state and modeled its inward-facing (IF) state using homology modeling. In addition,

they have used computational tools such as molecular dynamics to explore the conformational transition between the IF and OF states as well as binding and transport of the ions. The unique experimental-computational study reported here provides a hypothetical transport cycle with a detailed transport mechanism quantified energetically using computational techniques and kinetically using experimental techniques. The conclusion from both techniques is that the ion turnover frequency is extremely high that provides breast cancer cells with a highly efficient base loading mechanism resulting in their survival advantage. This work is of significance to the field it provides a full picture of the transport cycle of the NBCn1 transporter at the molecular detail. However, there are many technical issues with both the methodology and the presentation of the data particularly on the computational side of the story that makes it difficult to judge its reliability. Here I will go over some of these issues.

1. The presentation: It is hard to follow the logic of the manuscript since the information particularly on the computational side is scattered and incomplete. If one only read the Results section, it is impossible to know what computational methods the authors have done and need to go back and forth between the methods section and the results section to get an idea but even then, the information is still incomplete.

1.a. For instance, within the results section, the computational portion of the study starts with the statement that: "The IF NBCn1 structural model we generated superposed with the OF NBCn1 structure is shown in Supplementary Fig. S4a,b." It is not clear at all how the model is generated as there are many ways of doing that. Once jumping to the methods section, one can find "The IF state of NBCn1-B was built with the Swiss-Model server, using the recently resolved cryoEM structure of AE2 with bound Cl (PDB ID 8GV9)." This is the first appearance of AE2 and the text expects the reader to know what it stands for and why it is used for homology modeling. Also, it is never stated that the approach taken is homology modeling. Someone familiar with the Swiss-Model server will know this is the homology modeling approach and will guess the AE2 is a homologous protein but the manuscript does not actually state any of these.

1.b. A more serious example is the next part of the results section where the SLICS simulations are discussed. In that section, we read: "Average density maps from 1 μ s molecular dynamics (MD) simulations of apo-NBCn1 in OF and IF conformations (three replicas per state) in solution with the physiological relevant ions (0.75 M NaCl 195 + 0.375 M NaHCO₃ + 0.375 M Na₂CO₃) show similar permeation behavior of the cations and anions as the SILCS simulations (Fig. 3b)." It is not clear where we should find the details of these 1 μ s MD simulations. It sounds like there should be 6 (3 repeats x 2 states) simulations, each 1 μ s. However, in the methods section, within the SLICS methodology, one reads: "In total, 100 cycles of GCMC/MD comprising of 200,000 GCMC steps and 1 ns of MD simulation were performed, yielding a total of 1 μ s MD data for the NBCn1 OF and IF conformations." This implies the aggregate simulation time is perhaps 1 μ s but still it is not clear at all how as it only talks about (100 cycles x 1 ns = 100 ns) per systems.

1.c. The computational part of the results section is very unclear. It is almost never stated what methodology is used and one needs to go to the methods section to figure that out and even there, the information is not clear. Rather than expressing the actual sampling method used--this should be done in the results section--the authors have only mentioned the tool used and the reader needs to be familiar with the tool or look it up to figure out what the authors have done.

2. The analysis: As I explained above, the computational methodology used is very ambiguous and what makes this more challenging is that the analysis of the data generated is almost never provided. We never see any RMSD plot or any time series to determine whether any of these simulations are reliable or not. A very typical simulation both for the cryo-EM model and the homology model would have been a relatively long MD simulation to determine whether these simulations converge and whether these structures are stable. It looks like 200 ns simulations are performed that are not long enough for a system of this size. We are never told how the RMSD or RMSF behaves during these simulations. It is, for instance, very likely that during these simulations, specially the homology model one, the structure deviates significantly from the initial model. Given the fast transport cycle, it is actually plausible to assume one may observe early stages of transition within MD simulations. However, without providing the data, it is impossible to judge the quality of the homology model or even the cryo-EM model based on the MD simulations. More seriously, the free energy calculations does whether for substrate translocation or protein conformational transition or substrate binding free energy are simply provided as facts without any details on the convergence or error estimates. It is very easy and common for a free energy calculation simulation to get wrong numbers due to lack of convergence or the wrong use of methodology. There are various free energy calculation simulations run in this study that I cannot evaluate their reliability since I am not provided with any information. Do you observe any exchange in REMD simulations at all? Do you observe overlap between neighboring windows?

3. Why some simulations use cholesterol and some simulations are based on pure POPC? Why some simulations use Gromacs and some use NAMD? Why the energy unit sometimes kJ and sometimes kcal?

4. The calculated absolute binding free energies are extremely high (-73 to -93 kcal/mol) such that their validity is clearly questionable and the calculated state transition barriers are extremely low (under -3 kcal/mol). Such small barriers are not consistent even with the 15,000 1/s TOR.

Version 1:

Reviewer comments:

Reviewer #1

(Remarks to the Author)

The authors have shown more details of the ligands binding site at various contour levels to show the ion density is more significant than a water molecule. Also the authors have calculated Absolute free energy of binding (ΔG) and shown that the transporters have much stronger stabilization of $2\text{Na}^+-\text{CO}_3^{2-}$ than $\text{Na}^+-\text{HCO}_3^-$. I noticed reviewer 2# have also raised question about very negative free energy, and the authors have used the same calculation methods to show that free energy value calculated with $\text{Na}^+-\text{HCO}_3^-$ is much less than $2\text{Na}^+-\text{CO}_3^{2-}$, at least .

Overall, I have less concern about the current working model and the authors have also pointed out that they "cannot definitively rule out a $\text{Na}^+-\text{HCO}_3^-$ transport mode" which is fair and allow others' following research, which is good for the area.

I find the revised manuscript suitable for publication in nature communications with the high relevance for the transporter, cryoEM and breast cancer communities.

Reviewer #2

(Remarks to the Author)

The authors have now addressed my questions to a great extent.

We thank the Reviewers and Editor for the careful reviews and the detailed and very constructive comments. We have addressed each of the reviewer's comments and the manuscript has been revised as requested.

Reviewer #1

The manuscript submitted by Weiguang Wang et.al describes the first high resolution CryoEM structure of human NBCn1/SLC4A7 protein in complex with putative transporting ligands. NBCn1 is a highly important research target as the protein is crucial for breast cancer cells survival at acidic environment caused by aerobic respiration and lactic acid production. The authors have suggested new transporting ligands of the protein, unlike other close SLC4 members that transport HCO₃⁻, NBCn1 is actually transporting CO₃²⁻ together with 2 Na⁺, adding up to an electroneutral transporter.

This new mechanism is used to explain why NBCn1, not other SLC4 members are so relevant to the dramatic lower extracellular pH of breast tumor cell. The authors have done mutagenesis assay, MD simulation and thermodynamic calculation to show that the transporting ligand is suitable for base loading into the cancer cell at near physiological conditions. Also the authors have measured the turnover rate of the transporter and shown NBCn1 to be one of the fastest symporters reported to date.

Overall the manuscript is well written, the result is novel, and would provide new insight that is relevant to many both in transporter structure community and breast cancer research community. But I do have concerns about the new mechanism as the authors have not discussed about the possibility AT ALL that NBCn1 could still be a Na⁺ and HCO₃⁻ co-transporter. As the authors have shown, the HCO₃⁻ / CO₃²⁻ concentration ratio is 24.3/0.0367 ~ 600 fold at pH7.4 and 6.09/0.00232 ~2600 folds at pH6.8, which means that the transporter have a much higher probability to encounter HCO₃⁻ ion at physiological conditions, how could this happen if NBCn1 is the fastest transporter?

It is not very easy to tell HCO₃⁻ and CO₃²⁻ from CryoEM density as both ligands have the same triangle shape. Further, it is very important to show the putative ion density in the CryoEM density with different contour level as it's usually not to easy to tell sodium ion with water, either. Could there be possibility that one of the proposed ion densities is Na and the other one is water? That make the transporter a water dependent Na⁺ and HCO₃⁻ co-transporter, which is the ligands previously suggested by most of the researcher with less change comparing to the current mechanism suggested by the authors.

I would like the author to consider that possibility and compare with their current analysis why they prefer one from another as I could not see any violation of a water dependent Na⁺ and HCO₃⁻ co-transporter with the reasoning they have made in the manuscript, both MD and thermodynamic calculation.

If the authors could clear my concern, I would be very thankful and suggest the manuscript to be accepted in NC, but currently I do worry that the author have come to a too bold hypothesis with limited information from CryoEM density, which could be easily misinterpreted.

Per the reviewer's request, we examined the densities in the cryoEM map at multiple contour levels. The contour data is now shown in revised Supplementary Fig. S4. At the S1 and S2 sites, the putative Na⁺ densities persist across a wide range of contour levels and maintain a strong signal intensity compared to the density of putative CO₃²⁻ consistent with Na⁺ rather than H₂O. We would also like to point out that despite the physiologically higher concentration of HCO₃⁻ compared to CO₃²⁻, the related Na⁺-coupled SLC4 transporters NBCe1 and NDCBE are also thought to transport CO₃²⁻ (J Amer Soc Nephrol, 34(1):p 8-13, 2023; J Am Soc Nephrol, 34(1):40-54, 2023; Nat Commun 2021 Sep 28;12(1):5690). In the revised manuscript we have included free energy of binding calculations for IF and OF NBCn1 bound to Na⁺-HCO₃⁻ for comparison (revised Supplementary Table S2). The free energy of binding calculations data also support the preferential binding of 2Na⁺-CO₃²⁻ versus Na⁺-HCO₃⁻. Although we cannot definitively rule out a Na⁺-HCO₃⁻ transport mode, our findings are most compatible with NBCn1 mediating the highly efficient transport of two Na⁺ and a CO₃²⁻. We address these considerations in the revised manuscript as requested.

Reviewer #2

The manuscript by Wang et al. reports a combined experimental and computational study of the structural dynamics of NBCn1 transporter that helps with the survival of the breast cancer cells. They have determined a cryoEM structure for this transporter in the outward-facing (OF) state and modeled its inward-facing (IF) state using homology modeling. In addition, they have used computational tools such as molecular dynamics to explore the conformational transition between the IF and OF states as well as binding and transport of the ions. The unique experimental-computational study reported here provides a hypothetical transport cycle with a detailed transport mechanism quantified energetically using computational techniques and kinetically using experimental techniques. The conclusion from both techniques is that the ion turnover frequency is extremely high that provides breast cancer cells with a highly efficient base loading mechanism resulting in their survival advantage. This work is of significance to the field it provides a full picture of the transport cycle of the NBCn1 transporter at the molecular detail. However, there are many technical issues with both the methodology and the presentation of the data particularly on the computational side of the story that makes it difficult to judge its reliability. Here I will go over some of these issues.

1. The presentation: It is hard to follow the logic of the manuscript since the information particularly on the computational side is scattered and incomplete. If one only read the Results section, it is impossible to know what computational methods the authors have done and need to go back and forth between the methods section and the results section to get an idea but even then, the information is still incomplete.

We have modified the text as requested both in the Results and the Methods sections.

1.a. For instance, within the results section, the computational portion of the study starts with the statement that: "The IF NBCn1 structural model we generated superposed with the OF NBCn1 structure is shown in Supplementary Fig. S4a,b." It is not clear at all how the model is generated as there are many ways of doing that. Once jumping to the methods section, one can

find "The IF state of NBCn1-B was built with the Swiss-Model server, using the recently resolved cryoEM structure of AE2 with bound Cl (PDB ID 8GV9)." This is the first appearance of AE2 and the text expects the reader to know what it stands for and why it is used for homology modeling. Also, it is never stated that the approach taken is homology modeling. Someone familiar with the Swiss-Model server will know this is the homology modeling approach and will guess the AE2 is a homologous protein but the manuscript does not actually state any of these.

The text has been modified as requested indicating that our NBCn1 IF homology model was generated using the homologous SLC4 AE2 anion exchanger cryoEM IF structure as a template (see the changes in the "IF homology model of NBCn1" section in the Results and in the "Homology Modeling of the NBCn1 IF conformation" section in Methods.

1.b. A more serious example is the next part of the results section where the SILCS simulations are discussed. In that section, we read: "Average density maps from 1 μ s molecular dynamics (MD) simulations of apo-NBCn1 in OF and IF conformations (three replicas per state) in solution with the physiological relevant ions (0.75 M NaCl 195 + 0.375 M NaHCO₃ + 0.375 M Na₂CO₃) show similar permeation behavior of the cations and anions as the SILCS simulations (Fig. 3b)." It is not clear where we should find the details of these 1 μ s MD simulations. It sounds like there should be 6 (3 repeats x 2 states) simulations, each 1 μ s. However, in the methods section, within the SILCS methodology, one reads: "In total, 100 cycles of GCMC/MD comprising of 200,000 GCMC steps and 1 ns of MD simulation were performed, yielding a total of 1 μ s MD data for the NBCn1 OF and IF conformations." This implies the aggregate simulation time is perhaps 1 μ s but still it is not clear at all how as it only talks about (100 cycles x 1 ns = 100 ns) per systems.

We apologize for the confusion due to the lack of provided details. The Results section where we initially presented the SILCS results also contained results from separate 1 μ s long unrestrained MD simulations done with the Anton2 supercomputer (3 x 1 μ s MD replicas for each state (IF and OF) for a total of 6 x 1 μ s MD simulations). These simulations were used to probe the behavior of the physiologically relevant cations and anions (Na⁺, Cl⁻, HCO₃⁻ and CO₃²⁻) as opposed to the SILCS probes (methylammonium and acetate ions). The SILCS simulations are done in 10 replicas, with 100 ns MD sampling per replica, leading to a total of 1 μ s MD sampling. Accordingly, we have modified the text throughout the manuscript to provide the missing details and we have changed the headings of the relevant Results and Methodology sections to make the logical connections in the text clearer and easier to understand (see the changes in the section "Ion permeation dynamics in the NBCn1 OF and IF conformations from SILCS and unrestrained 1 μ s MD simulations" in Results and the sections "OF and IF conformation SILCS calculations" and "Unrestrained 200 ns MD simulations of OF and IF NBCn1 with 2Na⁺-CO₃²⁻ and 1 μ s MD simulations of OF and IF apo-NBCn1" in Methods).

1.c. The computational part of the results section is very unclear. It is almost never stated what methodology is used and one needs to go to the methods section to figure that out and even there, the information is not clear. Rather than expressing the actual sampling method used--this should be done in the results section--the authors have only mentioned the tool used and the reader needs to be familiar with the tool or look it up to figure out what the authors have done.

As suggested, we have now provided the methodological details in the relevant Results sections and have changed the headings of some Results sections to make their contents clearer (see

changes in sections “IF homology model of NBCn1”, “Ion permeation dynamics in the NBCn1 OF and IF conformations from SILCS and unrestrained 1 μ s MD simulations”, and “Position-restrained and unrestrained 200 ns MD simulations of NBCn1 for identification and refinement of ion binding sites in the OF and IF conformational states”).

2.The analysis: As I explained above, the computational methodology used is very ambiguous and what makes this more challenging is that the analysis of the data generated is almost never provided. We never see any RMSD plot or any time series to determine whether any of these simulations are reliable or not. A very typical simulation both for the cry-EM model and the homology model would have been a relatively long MD simulation to determine whether these simulations converge and whether these structures are stable. It looks like 200 ns simulations are performed that are not long enough for a system of this size. We are never told how the RMSD or RMSF behaves during these simulations. It is, for instance, very likely that during these simulations, specially the homology model one, the structure deviates significantly from the initial model. Given the fast transport cycle, it is actually plausible to assume one may observe early stages of transition within MD simulations. However, without providing the data, it is impossible to judge the quality of the homology model or even the cryo-EM model based on the MD simulations.

The RMSD plots for the Ca atoms of the transmembrane helices (with loops excluded from the RMSD calculation) evaluated from the six 1 μ s long unrestrained MD trajectories are now shown in Supplementary Fig. S15 and can provide the necessary information about the structural stability of the OF and IF NBCn1 states. For the most part, the RMSDs indicate structural convergence within the simulated 1 μ s trajectories with $\sim 2\text{\AA}$ RMSD deviation from the initial geometries in both states. Potential structural instabilities with RMSD approaching 3\AA are observed only in replica 3 in the IF state where a concerted motion of TM3 and TM10, consistent with initial stages of elevator IF \leftrightarrow OF transport leads to smooth increase of the RMSD before a plateau is reached. The motion of the TM3 and TM10, however, did not lead to the occlusion of the IF cavity or to the opening of the OF cavity at this stage and water and ion access to the protein center from the intracellular side was not obstructed in this putative early intermediate state. This information has been added as well in the “Unrestrained 200 ns MD simulations of OF and IF NBCn1 with a $2\text{Na}^+\text{-CO}_3^{2-}$ ion load and 1 μ s MD simulations of OF and IF apo-NBCn1” section in Methods.

The 200 ns MD simulations with a $2\text{Na}^+\text{-CO}_3^{2-}$ ion load were done for optimization and refinement of putative cation and anion binding sites in the vicinity of the protein center. We chose a 200 ns trajectory length for these simulations based on previous work in other SLC4 transporters (J Biol Chem. 2021;296:100724; Nat Commun. 2021;12:5690). They were long enough to assess ion dynamics at the protein center and to observe prolonged binding or ion dissociation in the wide and very well-hydrated OF and IF cavities, but they were short enough to avoid major conformation changes that could potentially interfere with the ion dynamics. The data in the previous Supplementary Table 2 in the original manuscript is now presented in a new Supplementary Fig. S6 which shows time series plots for the ions involved in these simulations where ion binding, dissociation and entry in the vicinity of sites S1/2^{MD}(OF/IF) can be tracked throughout the MD trajectory. Accordingly, changes have been made in the “Position-restrained and unrestrained 200 ns MD simulations of NBCn1 for identification and refinement of ion binding sites in the OF and IF conformational states” section in Results.

2.a. More seriously, the free energy calculations does whether for substrate translocation or protein conformational transition or substrate binding free energy are simply provided as facts without any details on the convergence or error estimates. It is very easy and common for a free energy calculation simulation to get wrong numbers due to lack of convergence or the wrong use of methodology. There are various free energy calculation simulations run in this study that I cannot evaluate their reliability since I am not provided with any information. Do you observe any exchange in REMD simulations at all? Do you observe overlap between neighboring windows?

In Supplementary Table 2 we now present free energy values calculated from the last 4 ns of the free energy calculations with standard deviation. The small standard deviations indicate convergence of the evaluated free energies. All replicas during the simulations were involved in exchanges with neighboring replicas. We have included information on the convergence and replica exchange in the “Absolute free energy of binding calculations” section in Methods.

The free energy of binding simulations are the only free energy simulations we have done in the manuscript. The free energy plots used for discussing ion permeation and the OF ↔ IF transition were extracted from population analysis based on position-restrained MD simulations or clustering of unrestrained MD trajectories of a string of Climber intermediate states. As such, they are more useful for qualitative (e.g. position of maxima and minima that can indicate ion binding sites and barriers along the permeation paths, or identification of a structural cluster representing an occluded intermediate state) rather than quantitative analysis. The numerical values of the mapped barriers should also be viewed more qualitatively (e.g. to address whether ion motion along the permeation pathways encounter numerous prohibitive barriers).

3. Why some simulations use cholesterol and some simulations are based on pure POPC? Why some simulations use Gromacs and some use NAMD? Why the energy unit sometimes kJ and sometimes kcal?

*The choice of specific program has been made because it was either necessary for the used method or would be the most optimal choice for resource usage. For instance, the free energy simulations method we use (λ -REMD FEP) is implemented in NAMD. For consistency, we also used NAMD for the 200 ns MD simulations of the OF and IF NBCn1 states from which we obtained refined geometries for the protein with a $2\text{Na}^+\text{-CO}_3^{2-}$ ion load bound at the protein center. The available implementations of SILCS make use of Gromacs and work with a membrane that has both POPC and cholesterol. We have opted for pure POPC bilayers for our other simulations, both for consistency with earlier published data on other SLC4 transporters (*J Biol Chem.* 2021;296:100724; *Nat Commun.* 2021;12(1):5690) and for simplification of the system size and complexity. The 1 μs long unrestrained MD simulations were done on the Anton2 supercomputer using the local specialized software since this was the fastest available to us option for them. The latest Gromacs versions are considerably faster than NAMD 2.14 on the various in-house and Digital Research Alliance of Canada clusters we have access to, hence we opted to use them for calculations that required a lot of sampling in multiple small replicas, such as the position-restrained MD simulations or the MD simulations with Climber states. We have throughout our calculations compared the results of these different simulation methodologies and the identified ion binding sites and permeation pathways were consistent across the various methods used.*

The energy kJ-based energy units are used in the text only in the Methods section where the software specifically requires kJ-based units as input. NAMD directly produced the absolute free energy of binding results in kcal/mol. Consequently, we decided to show the results from free energy calculations or population analysis in kcal/mol since such units are widely used for protein-substrate binding or energetics of conformational transitions. For consistency, we have added in brackets the force constant/force in kcal-based units in the Methods section.

4. The calculated absolute binding free energies are extremely high (-73 to -93 kcal/mol) such that their validity is clearly questionable and the calculated state transition barriers are extremely low (under -3 kcal/mol). Such small barriers are not consistent even with the 15,000 1/s TOR.

We agree that the absolute binding free energy values in Supplementary Table 2 for $2\text{Na}^+\text{-CO}_3^{2-}$ binding are very negative. However, we have observed similar very negative values for absolute or relative binding free energies of CO_3^{2-} in SLC4 proteins with other methods such as Poisson-Boltzman based ones (MMPBSA or Charmm-gui's PBEQ) when Na^+ ions are present in the binding site. For example, PBEQ calculations of OF and IF NBCn1 with $2\text{Na}^+\text{-CO}_3^{2-}$, done for comparison to the λ -REMD FEP values reported in Supplementary Table S2, yield -44.37 and -60.08 kcal/mol for CO_3^{2-} binding to the $\text{S1}^{\text{MD}}(\text{OF})$ and $\text{S1}^{\text{MD}}(\text{IF})$ site of NBCn1 containing 2 Na^+ ions. Very negative free energies of binding when Na^+ and CO_3^{2-} are present together in the binding sites have been obtained even with quantum chemical methods, which do not rely on ad-hoc force fields, such as DFT. As an example, in another member of the SLC4 family, NBCe1, DFT calculations for relative binding free energy between toy models of the S1 site of $\text{NBCe1}+\text{HCO}_3^-$ (or Cl^-) and $\text{NBCe1}+\text{Na}^++\text{CO}_3^{2-}$ yielded relative binding free energies below -30 kcal/mol or -145 kcal/mol in implicit water or protein environments (<https://ucalgary.scholaris.ca/items/5810d048-a11d-4a5d-81c6-2fc0847d05cf>) showing significant stabilization of the $\text{Na}^+\text{-CO}_3^{2-}$ pair in the binding pocket. As mentioned in our response to Reviewer 1, we have included in the revised manuscript, absolute binding free energy values for $\text{Na}^+\text{-HCO}_3^-$ (revised Supplementary Table S2) for relative comparison. The absolute binding free energy values for HCO_3^- obtained with the same approach were much less negative reflecting the -1 charge of HCO_3^- and the presence of only one Na^+ ion in the binding site: -2.98 ± 0.78 for $\text{S1}^{\text{MD}}(\text{OF})$ and -10.94 ± 0.67 for $\text{S1}^{\text{MD}}(\text{IF})$ sites. These findings suggest that CO_3^{2-} is the preferred NBCn1 substrate.

Regarding the low calculated state transition barriers: the aim of the clustering analysis done on the OF \leftrightarrow IF transition was to extract an occluded intermediate state. This point is now stated clearly in the revised manuscript in the Results ("Computational modeling of the OF \leftrightarrow IF transition and identification of an occluded apo intermediate state in the transport cycle"). In this case the provided free energy landscapes are used in a qualitative manner for the identification of the occluded state, since they were obtained from limited MD sampling.

Additional Changes:

In addition to addressing the comments of Reviewer 1 and 2, the revised manuscript includes minor typographical/grammatical error corrections, supplementary figure numbering changes and referencing additions/numbering changes.